# Are Global Dependencies Necessary? Scalable Time Series Forecasting via Local Cross-Variate Modeling

**Kun Liu**[1], **Renjun Jia**[2], **Ruifeng Yang**[1], **Xirui Zeng**[1], **Yuqi Liang**[3], **Cen Chen**[1]*
[1]East China Normal University, [2]Tongji University, [3]Seek Data Group, Emoney Inc.
`51265900028@stu.ecnu.edu.cn`, `cenchen@dase.ecnu.edu.cn`

## ABSTRACT

Effectively modeling cross-variate dependencies is a central, yet challenging, task in multivariate time series forecasting. While attention-based methods have advanced the state-of-the-art by capturing global cross-variate dependencies, their quadratic complexity with respect to the number of variates severely limits their scalability. In this work, we challenge the necessity of global dependency modeling. We posit, through both theoretical analysis and empirical evidence, that modeling local cross-variate interactions is not only sufficient but also more efficient for many dense dependency systems. Motivated by this core insight, we propose VPNet, a novel architecture that excels in both accuracy and efficiency. VPNet's design is founded on two key principles: a channelized reinterpretation of patch embeddings into a higher-level variate-patch field, and a specialized VarTCNBlock that operates upon it. Specifically, the model first employs a patch-level autoencoder to extract robust local representations. In a pivotal step, these representations are then re-conceptualized as a 2D field constructed over a "variates × patches" grid. The VarTCNBlock then applies depthwise 2D convolutions across this field to efficiently capture local spatio-temporal patterns (i.e., cross-variate and temporal dependencies simultaneously), followed by pointwise convolutions for feature mixing. This design ensures that the computational complexity scales linearly with the number of variates. Finally, variate-wise prediction heads map the refined historical patch representations to future ones, which are decoded back into the time domain. Extensive experiments demonstrate that VPNet not only achieves state-of-the-art performance across multiple benchmarks but also offers significant efficiency gains, establishing it as a superior and scalable solution for high-dimensional forecasting. Code is available at this repository: https://github.com/iuaku/VPNet/

## 1 INTRODUCTION

Multivariate time series forecasting is a cornerstone of data-driven decision-making, with critical applications spanning a wide range of domains from energy grid management and traffic flow prediction to meteorology and finance (Granger & Newbold, 2014; Martín et al., 2010; Qian et al., 2019; Chen et al., 2001; Yin et al., 2021; Wu et al., 2023b). A key technical challenge is modeling *cross-variate* dependencies: the complex, time-varying interactions among many co-evolving series (Zhang & Yan, 2023; Liu et al., 2024a). Effective modeling of these dependencies is crucial for accurate long-horizon forecasting in high-dimensional regimes.

Recent progress has been driven by channel-fusion architectures (Zhao & Shen, 2024), particularly Transformer-based designs that explicitly model global cross-variate interactions (e.g., iTransformer (Liu et al., 2024a)). These models attain strong predictive performance by searching for dependencies across all variates, but their expressivity comes at a steep computational and memory cost: the cost of channel-mixing attention grows quadratically with the number of variates, making such models impractical for systems with hundreds or thousands of variates. At the opposite end of the design spectrum, channel-independent models (including PatchTST (Nie et al., 2023), TimeMixer (Wang

---

*Corresponding Author

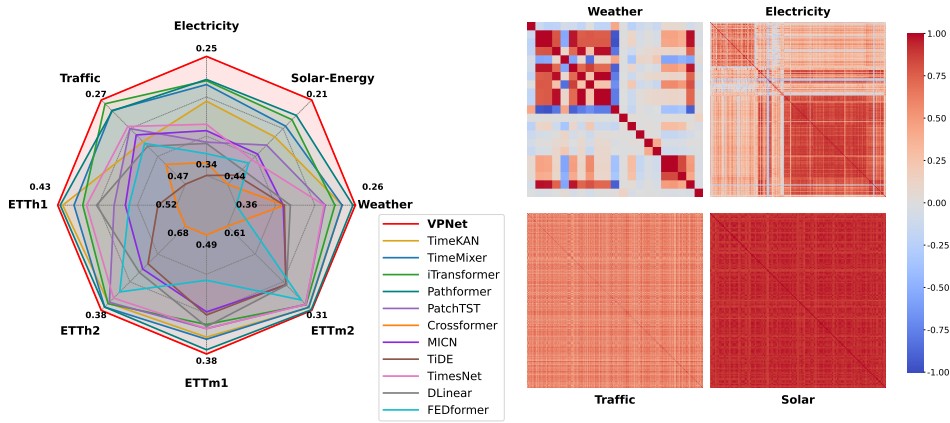

Figure 1: Benchmarking model performance on eight datasets (left) and variate correlation analysis on four high-dimensional datasets (right). For model performance, the average MAE across four prediction tasks on each dataset is used as the comparison result.

et al., 2024a), Dliner (Zeng et al., 2023)) are highly efficient, yet by construction they forgo explicit cross-variate modeling and therefore struggle to exploit important inter-series predictive signals.

This accuracy–scalability tension raises a fundamental question:

> *Is searching for global dependencies necessary for accurate forecasting in dense, high-dimensional systems?*

In response, we formulate the **Local Sufficiency Hypothesis**: in many real-world dense systems, the dependency graph is sufficiently rich that an appropriately chosen finite local neighborhood will almost surely capture the key signals required for prediction. This makes exhaustive global search not only computationally unnecessary but also potentially noise-amplifying.

We support this principle via two complementary pillars of evidence. First, we provide a probabilistic analysis showing that under realistic dense-dependency regimes, a bounded local neighborhood has a high probability of including informative neighbors (Theorem 1; Appendix C). Second, we corroborate this theoretical insight empirically. Figure 1 (right) displays correlation heatmaps for four high-dimensional benchmarks, which all exhibit the strong, dense inter-variate correlations that our hypothesis relies upon. This evidence collectively validates our focus on developing an architecture centered on local, rather than global, interactions.

Guided by this Local Sufficiency Hypothesis, we introduce **VPNet**, a principled architecture that attains strong predictive accuracy while scaling linearly with the number of variates. The model first employs a patch-level autoencoder, a technique proven effective in recent literature, to generate robust representations of local temporal patterns. Building on these representations, VPNet rests on two core ideas. (1) We reinterpret the patch embeddings as a higher-level *variate–patch field* organized as a 2D grid over *variates* and *temporal patches*. This representation enables the model to treat cross-variate and temporal structure jointly at a coarser, more robust abstraction level. (2) We design the *VarTCNBlock*, a lightweight module that applies efficient depthwise 2D convolutions over the variate–patch field to capture local spatio-temporal patterns. By restricting computation to local neighborhoods and using depthwise operations, VPNet achieves linear complexity with the number of variates. Finally, a variate-wise prediction head maps the refined historical patch representations to future ones, which are then mapped back to the time domain by the autoencoder's decoder. As previewed in 1 (left), this principled design allows VPNet to achieve state-of-the-art performance across eight forecasting benchmarks, consistently outperforming strong baselines while offering substantial efficiency gains (see 4.1 for full analysis).

We summarize our main contributions as follows:

- We formulate the Local Sufficiency Hypothesis, providing both theoretical and empirical evidence to challenge the necessity of global dependency modeling in dense systems.

- We introduce the variate-patch field and the VarTCNBlock, a novel representation and architecture that operationalize our hypothesis and enable the efficient capture of cross-variate dependencies with linear complexity.
- We establish new state-of-the-art results across eight diverse forecasting benchmarks, demonstrating that VPNet simultaneously achieves superior accuracy and linear scalability, effectively resolving the critical accuracy-efficiency trade-off.

## 2 PRELIMINARIES

This section fixes the notations and describes the patch-level overcomplete autoencoder used as input/output projection, so the Method section can focus on the novel components.

**Notation and problem formulation.** Let $X \in \mathbb{R}^{B \times L \times C}$ denote a minibatch of $B$ multivariate time series samples, where $L$ is the look-back window length and $C$ is the number of variates (channels). The forecasting objective is to predict the subsequent $S$ time steps, denoted by $Y \in \mathbb{R}^{B \times S \times C}$, from the history $X$. We denote the model by $f_\theta$ and its prediction by $\widehat{Y} = f_\theta(X)$.

Let the patch length be $p \in \mathbb{Z}_+$. For simplicity we assume $L$ is divisible by $p$ and define the number of non-overlapping patches $P = L \mid p$. For batch index $b \in \{1, \ldots, B\}$, variate index $c \in \{1, \ldots, C\}$, and patch index $i \in \{1, \ldots, P\}$, the $i$-th temporal patch is denoted as $x_{b,c,i} \in \mathbb{R}^p$.

We adopt the mean absolute error (MAE, $L_1$) as the prediction loss due to its robustness to outliers:

$$\mathcal{L}_{\text{pred}} \;=\; \frac{1}{B\,S\,C} \sum_{b=1}^{B} \sum_{t=1}^{S} \sum_{c=1}^{C} \left\| \widehat{Y}_{b,t,c} - Y_{b,t,c} \right\|_1. \tag{1}$$

**Patch-level overcomplete autoencoder.** To obtain robust, locally informative representations of non-stationary time series, we employ a patch-level overcomplete autoencoder (Liu et al., 2025b) as the input and output projection module. A shared encoder $\text{Enc} : \mathbb{R}^p \to \mathbb{R}^H$ and shared decoder $\text{Dec} : \mathbb{R}^H \to \mathbb{R}^p$ are defined by

$$e_{b,c,i} = \text{Enc}(x_{b,c,i}) \in \mathbb{R}^H, \qquad \tilde{x}_{b,c,i} = \text{Dec}(e_{b,c,i}) \in \mathbb{R}^p. \tag{2}$$

The encoder output is followed by Layer Normalization:

$$\bar{e}_{b,c,i} = \text{LayerNorm}(e_{b,c,i}) \in \mathbb{R}^H. \tag{3}$$

We typically choose an *overcomplete* latent dimension $H > p$ to provide redundant capacity for representing complex patch dynamics. The encoder and decoder are implemented as lightweight MLPs (e.g., two linear layers with nonlinearity) and their parameters are shared across variates and patches, yielding a universal patch basis. Applying the encoder + LayerNorm to every patch produces the initial embedding tensor $\mathbf{E} \in \mathbb{R}^{B \times C \times P \times H}$.

To enforce reconstruction fidelity we include an auxiliary reconstruction loss:

$$\mathcal{L}_{\text{rec}} \;=\; \frac{1}{B\,C\,P} \sum_{b=1}^{B} \sum_{c=1}^{C} \sum_{i=1}^{P} \left\| \tilde{x}_{b,c,i} - x_{b,c,i} \right\|_1. \tag{4}$$

The total training objective balances prediction and reconstruction:

$$\mathcal{L}_{\text{total}} \;=\; \mathcal{L}_{\text{pred}} \;+\; \mathcal{L}_{\text{rec}}, \tag{5}$$

## 3 METHOD

Global cross-variate dependency search incurs prohibitive computational and memory costs, while purely channel-independent models lack sufficient expressive power. Grounded in the *local-sufficiency hypothesis*, we introduce **VPNet** (Variate–Patch Network), a scalable architecture for multivariate time series forecasting that exploits localized structures without resorting to full global mixing. As illustrated in Figure 2, VPNet is built upon the patch-level overcomplete autoencoder described in Section 2 and follows a sequence-to-sequence paradigm operating on patch embeddings. Its core

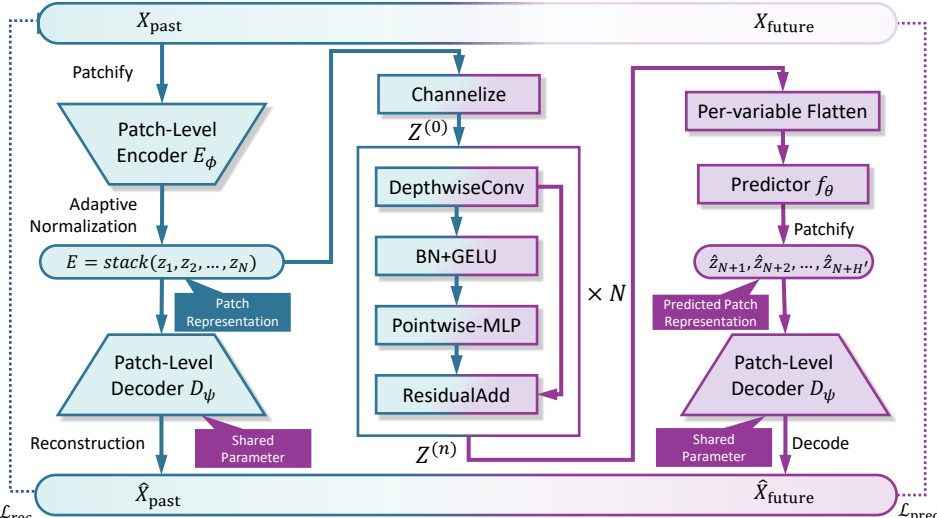

Figure 2: VPNet overall framework. Inputs are projected by the patch-level overcomplete autoencoder and channelization to produce $Z^{(0)}$, which is processed by stacked VarTCNBlocks and finally decoded to produce the forecast.

pipeline consists of four stages: (1) an *input projection* module that encodes raw series into patch embeddings via the patch autoencoder; (2) a *channelization* step that reinterprets patch embeddings as a variate–patch field to expose cross-variate structure; (3) a stack of *VarTCNBlocks* that progressively transform the channelized field through efficient local convolution; and (4) an *output projection* module that decodes the refined patch representations into the final forecast.

**Design Motivation: The Local Sufficiency Hypothesis.** A core design principle of VPNet is that, in dense systems where many variates are mutually informative, a small local neighborhood along the variate axis suffices with high probability to include an informative neighbor for any given target variate. We formalize this intuition in the following theorem; the detailed concentration proof is deferred to Appendix C.

**Theorem 3.1** (High-probability capture of informative neighbors). *Fix a target variate. Suppose that among the remaining $C-1$ variates, exactly $r$ variates belong to an* information set *$\mathcal{I} \subset \{1,\ldots,C\}$ (i.e., these $r$ variates are truly informative for predicting the target). Assume the variate ordering is a random permutation. Let $\mathcal{W}_k$ denote a contiguous window of width $k$ centered (or centered as close as boundary allows) on the target, and define the event*

$$\mathcal{E}_k = \{ |\mathcal{I} \cap \mathcal{W}_k| \geq 1 \}, \tag{6}$$

*meaning the window contains at least one informative variate. Then*

$$\mathscr{P}\mathfrak{r}[\mathcal{E}_k] \geq 1 - \exp\left(-\frac{kr}{C-1}\right). \tag{7}$$

*Sketch.* See Appendix C.2 for the full derivation. Intuitively, under random permutation the expected number of informative variates inside $\mathcal{W}_k$ is $\mu = kr/(C-1)$; a Chernoff/Poisson-style bound on the tail yields the stated exponential lower bound. □

**Corollary 3.1** (practical kernel selection). *To guarantee that $\mathcal{W}_k$ contains an informative variate with probability at least $1-\delta$, it suffices to choose*

$$k \geq \frac{C-1}{r} \ln\frac{1}{\delta}. \tag{8}$$

*This provides a direct, interpretable guideline for initializing the variate-axis kernel width $k$, which can then be fine-tuned empirically.*

## 3.1 STRUCTURE OVERVIEW

**Input projection.** Inspired by AdaPatch, we leverage a patch-level overcomplete autoencoder to handle potential distribution shifts and to extract local temporal patterns. Concretely, the input time series $X \in \mathbb{R}^{B \times L \times C}$ is partitioned into $P = L/p$ non-overlapping patches of length $p$. Denote the $i$-th patch of batch $b$ and variate $c$ by $x_{b,c,i} \in \mathbb{R}^p$. A shared encoder $\mathrm{Enc}(\cdot)$ maps each patch to a high-dimensional latent vector:

$$e_{b,c,i} \; = \; \mathrm{Enc}\big(x_{b,c,i}\big) \in \mathbb{R}^H, \qquad i = 1, \ldots, P. \tag{9}$$

Stacking these latents for all $b, c, i$ yields the initial patch representation tensor $\mathbf{E} \in \mathbb{R}^{B \times C \times P \times H}$, which serves as the input to the subsequent channelization.

**Channelization and the Variate-Patch Field.** This reinterpretation is central to our method and marks a conceptual departure from prior TCN-based models. As illustrated in the figure, while prior works such as ModernTCN and TimesNet treat the time series as a 2D input with a single channel or reshape it into a 2D plane based on periodicity to capture intra-series patterns, our approach innovatively treats each patch as a holistic unit to construct a high-dimensional Variate-Patch Field (VP-Field). Specifically, the initial representation $\mathbf{E}$ is permuted to create the channelized variate-patch field $\mathbf{Z}^{(0)}$:

$$\mathbf{Z}^{(0)} = \mathrm{Permute}(\mathbf{E}) \in \mathbb{R}^{B \times H \times C \times P}. \tag{10}$$

This permutation recasts the patch embedding dimension $H$ as the channel dimension for a 2D operator, while the variates $C$ and patches $P$ form a two-dimensional spatial grid we term the *variate-patch field*. By applying convolution directly on this high-level semantic field, we can simultaneously and efficiently capture both cross-variate and temporal dependencies.

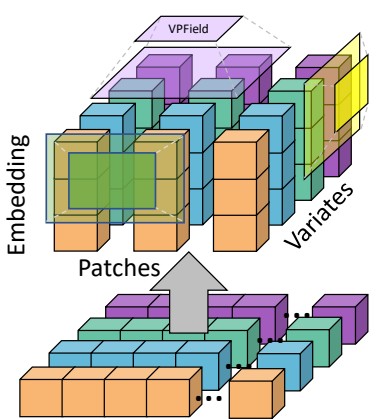

**VarTCNBlocks.** The channelized tensor $\mathbf{Z}^{(0)}$ is then processed by a stack of $N$ VarTCNBlocks, which iteratively refine the patch representations. The abstract forward propagation through the stack is defined as:

Figure 3: (a) Traditional Convolution (Yellow): Applying convolutional operations directly on the original time series. (b) Frequency-based 2D Convolution (Green): Reconstructing independent variates into a two-dimensional plane based on their frequency characteristics. (c) Our Method (Purple)

$$\mathbf{Z}^{(l+1)} \; = \; \mathrm{VarTCNBlock}\big(\mathbf{Z}^{(l)}\big), \qquad l = 0, \ldots, N - 1, \tag{11}$$

Each block operates on the channelized input $\mathbf{Z}^{(l)} \in \mathbb{R}^{B \times H \times C \times P}$ and produces an output of the same shape, progressively capturing more complex dependencies across both variates and patches.

**Output projection.** After $N$ blocks the stack produces the context-aware representation $Z^{(N)} \in \mathbb{R}^{B \times C \times P \times H}$. We apply a channel-independent prediction head $\mathrm{Head}(\cdot)$ to map each variate's historical patch sequence to future patch embeddings. Concretely, for each variate $c$ we flatten its history

$$u_{b,c} \; = \; \mathrm{vec}\big(Z^{(N)}_{b,c,:,:}\big) \in \mathbb{R}^{HP}, \tag{12}$$

and a shared per-variate MLP $\mathrm{Head} : \mathbb{R}^{HP} \to \mathbb{R}^{HP_p}$ produces predicted patch coefficients

$$\widehat{u}_{b,c} \; = \; \mathrm{Head}(u_{b,c}) \in \mathbb{R}^{HP_p}, \tag{13}$$

which we reshape into predicted future patch embeddings $\widehat{Z} \in \mathbb{R}^{B \times C \times P_p \times H}$ (here $P_p$ is the number of predicted patches and the forecast horizon is $S = P_p \cdot p$). Finally, the shared decoder $\mathrm{Dec}(\cdot)$ maps each predicted patch embedding back to the time domain and we concatenate these patch-level reconstructions to obtain the final forecast

$$\widehat{Y} \; = \; \mathrm{Concat}_{i=1}^{P_p} \mathrm{Dec}\big(\widehat{Z}_{:,:,i,:}\big) \in \mathbb{R}^{B \times S \times C}. \tag{14}$$

**Remarks.** The prediction head is *channel-independent* (shared across variates) but operates on per-variate flattened patch histories; this design preserves parameter efficiency while allowing each variate to leverage mixed information aggregated by the VarTCN stack. The decoder is reused from the autoencoder (weight sharing) to regularize predicted embeddings and improve reconstruction fidelity.

## 3.2 The VarTCNBlock

The VarTCNBlock is the core computational engine of VPNet, designed to efficiently process the channelized variate-patch field $\mathbf{Z}^{(l)} \in \mathbb{R}^{B \times H \times C \times P}$. Each block refines these representations by capturing dependencies across both variates and time, and its design is centered around a residual connection that wraps two main components: a depthwise convolution for local spatial mixing and a pointwise feed-forward network for feature transformation.

**Depthwise Convolution for Local Mixing:** Motivated by our probabilistic analysis that local interactions are sufficient for capturing salient signals in systems with dense dependencies, we employ a depthwise 2D convolution over the variate-patch field. Unlike standard convolution, a depthwise convolution applies a distinct 2D kernel $\mathbf{W}^{(h)} \in \mathbb{R}^{k_v \times k_p}$ to each input channel $\mathbf{Z}^{(l)}_{:,h,:,:}$ independently. This operation is formulated as:

$$\mathbf{Y}^{\mathrm{dw}}_h = \mathrm{DWConv2D}\big(\mathbf{Z}^{(l)}_{:,h,:,:}, \mathbf{W}^{(h)}\big), \quad h = 1, \ldots, H. \tag{15}$$

This step effectively aggregates information from a local neighborhood of size $k_v \times k_p$ on the variate-patch field, explicitly modeling temporally-local cross-variate dependencies. Crucially, as advocated in our introduction, this is achieved with a computational cost and parameter count that scale *linearly* with the number of variates $C$, making it exceptionally suitable for high-dimensional forecasting.

**Pointwise Feed-Forward Network for Feature Transformation:** Following the spatial mixing, the resulting features $\mathbf{Y}^{\mathrm{dw}}$ are passed through a normalization layer and an activation function before being processed by a feed-forward network (FFN). The FFN is implemented with pointwise ($1 \times 1$) convolutions and serves to mix information across the feature channels at each position on the variate-patch field. It follows an inverted bottleneck structure, first expanding the channel dimension by a factor of $r_{\mathrm{ff}}$ and then projecting it back. The full sequence of operations is:

$$\mathbf{Y}^{\mathrm{act}} = \mathrm{GELU}\big(\mathrm{BN}(\mathbf{Y}^{\mathrm{dw}})\big), \quad \mathbf{Y}^{\mathrm{ffn}} = \mathrm{FFN}(\mathbf{Y}^{\mathrm{act}}). \tag{16}$$

The FFN module itself consists of two pointwise convolutions, activations, and dropout for regularization. The output of this entire sequence is then added to the block's original input via a residual connection:

$$\mathbf{Z}^{(l+1)} = \mathbf{Z}^{(l)} + \mathbf{Y}^{\mathrm{ffn}}. \tag{17}$$

This residual design is essential for stabilizing the training of deep models by allowing gradients to flow more freely.

## 4 Experiments

In this section, we conduct a comprehensive empirical evaluation to validate the efficacy and efficiency of VPNet. We begin by benchmarking VPNet against a diverse suite of state-of-the-art models on several public datasets to establish its overall performance (Section 4.2). Subsequently, we perform a series of detailed ablation studies to dissect the model and verify the contributions of its core design principles, particularly the local cross-variate convolution mechanism (Section 4.3). Finally, we analyze the model's practical properties, focusing on its computational and memory efficiency (Section 4.5).

### 4.1 Experimental Setup

**Datasets.** We evaluate VPNet on eight widely-used public benchmarks for long-term time series forecasting: *Weather*, *Traffic*, *Electricity*, *Solar-Energy*, and four *ETT* datasets (ETTh1, ETTh2, ETTm1, ETTm2). Detailed statistics for each dataset are provided in Appendix B. We particularly focus on the high-dimensional datasets (*Electricity*, *Traffic*) which contain hundreds of variates, making them ideal for assessing the model's ability to handle cross-variate dependencies. All datasets are partitioned into training, validation, and testing sets following a 6:2:2 ratio for ETT datasets and 7:1:2 for the others. This aligns with prior benchmarks set by (Zhou et al., 2021; Liu et al., 2022).

**Baseline Details.** To provide a robust comparative analysis, we evaluate VPNet against a carefully curated collection of strong baselines that together represent the principal modelling paradigms for long horizon multivariate time series forecasting. This collection comprises KAN based models

exemplified by TimeKAN (2025), MLP centric architectures exemplified by TimeMixer (2024a), TiDE (2023) and DLinear (2023), Transformer variants designed for long sequence forecasting exemplified by iTransformer (2024a), PatchTST (2023), Pathformer (2024), Crossformer (2023) and FEDformer (2022), and convolutional approaches exemplified by MICN (2023) and TimesNet (2023a). Together these baselines span complementary inductive biases and computational tradeoffs, thereby providing a stringent benchmark for assessing VPNet's capacity to capture interactions across variates and to scale to high dimensional settings.

**Implementation Details.** All experiments were implemented in PyTorch and conducted on a single NVIDIA A100 40GB GPU. Following standard long-term forecasting protocols, we use a fixed input sequence length of $L = 96$ to predict future horizons of $S \in \{96, 192, 336, 720\}$. For a comprehensive list of all model hyperparameters and training configurations for each dataset, please refer to Appendix B.

## 4.2 MAIN RESULTS ON LONG-TERM FORECASTING

Table 1 reports the long-term forecasting results, with averages summarized in the main paper for clarity. VPNet achieves the best overall performance, outperforming all baselines on most datasets and metrics.

On high-dimensional datasets (*Weather*, *Solar-Energy*, *Electricity*, and *Traffic*), where cross-variate dependencies are dense, VPNet demonstrates substantial gains. For example, on *Electricity*, it reduces MSE by **9.0%** compared with iTransformer (0.162 vs. 0.178), and on *Traffic*, it achieves a **26%** improvement over TimeKAN (0.421 vs. 0.572). These results strongly support our proposed Local Sufficiency Hypothesis.

On the *ETT* benchmarks with lower dimensionality, VPNet remains competitive, frequently ranking first or second. For instance, it outperforms all baselines on *ETTm2* and *ETTh2*, and is only marginally behind TimeKAN on *ETTh1*. This consistency across both high-dimensional and low-dimensional regimes highlights the robustness and general applicability of VPNet without requiring dataset-specific adaptations.

Table 1: Long-term forecasting results. We average the results across 4 prediction lengths: $\{96, 192, 336, 720\}$. The best performance is highlighted in **red**, and the second-best is underlined. Full results can be found in Appendix K.

| Models | VPNet (Ours) | | TimeKAN (2025) | | TimeMixer (2024a) | | iTransformer (2024a) | | Pathformer (2024) | | PatchTST (2023) | | Crossformer (2023) | | MICN (2023) | | TiDE (2023) | | TimesNet (2023a) | | DLinear (2023) | | FEDformer (2022) | |
|---|---|---|---|---|---|---|---|---|---|---|---|---|---|---|---|---|---|---|---|---|---|---|---|---|
| Metric | MSE | MAE | MSE | MAE | MSE | MAE | MSE | MAE | MSE | MAE | MSE | MAE | MSE | MAE | MSE | MAE | MSE | MAE | MSE | MAE | MSE | MAE | MSE | MAE |
| Weather | **0.238** | **0.261** | 0.243 | 0.272 | 0.240 | 0.272 | 0.258 | 0.278 | 0.239 | 0.263 | 0.265 | 0.286 | 0.264 | 0.320 | 0.268 | 0.321 | 0.271 | 0.320 | 0.259 | 0.287 | 0.265 | 0.315 | 0.309 | 0.360 |
| Solar-Energy | **0.204** | **0.207** | 0.276 | 0.310 | 0.216 | 0.280 | 0.233 | 0.262 | 0.241 | 0.250 | 0.287 | 0.333 | 0.406 | 0.442 | 0.283 | 0.358 | 0.347 | 0.417 | 0.403 | 0.374 | 0.330 | 0.401 | 0.328 | 0.383 |
| Electricity | **0.162** | **0.251** | 0.197 | 0.286 | 0.182 | 0.273 | 0.178 | 0.270 | 0.182 | 0.269 | 0.216 | 0.318 | 0.244 | 0.334 | 0.196 | 0.309 | 0.252 | 0.344 | 0.193 | 0.304 | 0.225 | 0.319 | 0.214 | 0.327 |
| Traffic | **0.421** | **0.273** | 0.572 | 0.372 | 0.485 | 0.298 | 0.428 | 0.282 | 0.501 | 0.299 | 0.529 | 0.341 | 0.667 | 0.426 | 0.593 | 0.356 | 0.761 | 0.473 | 0.620 | 0.336 | 0.625 | 0.383 | 0.610 | 0.376 |
| ETTh1 | 0.434 | **0.427** | **0.426** | 0.431 | 0.447 | 0.440 | 0.454 | 0.447 | 0.455 | 0.429 | 0.507 | 0.472 | 0.529 | 0.522 | 0.475 | 0.481 | 0.541 | 0.507 | 0.458 | 0.450 | 0.461 | 0.458 | 0.498 | 0.484 |
| ETTh2 | **0.356** | **0.383** | 0.391 | 0.409 | 0.365 | 0.395 | 0.383 | 0.407 | 0.374 | 0.395 | 0.391 | 0.412 | 0.942 | 0.684 | 0.574 | 0.531 | 0.611 | 0.550 | 0.414 | 0.427 | 0.563 | 0.519 | 0.437 | 0.449 |
| ETTm1 | **0.376** | **0.382** | 0.386 | 0.398 | 0.381 | 0.396 | 0.410 | 0.410 | 0.382 | 0.386 | 0.402 | 0.406 | 0.513 | 0.495 | 0.423 | 0.422 | 0.419 | 0.419 | 0.400 | 0.406 | 0.404 | 0.408 | 0.448 | 0.452 |
| ETTm2 | **0.270** | **0.312** | 0.277 | 0.322 | 0.275 | 0.323 | 0.288 | 0.332 | 0.271 | 0.314 | 0.290 | 0.334 | 0.757 | 0.611 | 0.353 | 0.402 | 0.358 | 0.404 | 0.291 | 0.333 | 0.354 | 0.402 | 0.305 | 0.349 |

## 4.3 ABLATION STUDIES

To dissect our model and validate its core design principles, we conduct a series of targeted ablation studies.

**Effectiveness of Cross-variate Convolution.** To quantify the impact of our local cross-variate mechanism, we conduct an ablation on the kernel size $k_v$. We evaluate the model with a range of kernel sizes: $k_v \in \{1, 3, 7, 17, 27\}$, where $k_v = 1$ represents the channel-independent baseline. As shown in Table 2, performance improves dramatically when moving from $k_v = 1$ to $k_v = 3$, confirming that local variate mixing is crucial. However, we observe that further increasing the kernel size to 7, 17, and beyond offers diminishing returns and can even slightly degrade performance. This result provides strong empirical validation for our local-sufficiency hypothesis: for dense dependency systems, a compact local receptive field across variates already contains sufficient auxiliary predictive

signals. attempting to model wider, quasi-global interactions, validating the probabilistic motivation outlined in our introduction.

Table 2: Ablation study on the cross-variate kernel size $k_v$. We report the average MSE over all prediction lengths for each benchmark. The case $k_v = 1$ is the channel-independent baseline. Lower is better.

| VPNET | ETTh1 | ETTh2 | ETTm1 | ETTm2 | ECL | Traffic | Weather | Solar |
|---|---|---|---|---|---|---|---|---|
| $k_v = 1$ | 0.437 | 0.362 | 0.375 | 0.282 | 0.184 | 0.443 | 0.254 | 0.224 |
| $k_v = 3$ | 0.435 | 0.362 | 0.378 | 0.275 | 0.171 | 0.431 | 0.250 | 0.203 |
| $k_v = 7$ | 0.434 | 0.355 | 0.375 | 0.273 | 0.167 | 0.435 | 0.248 | 0.204 |
| $k_v = 17$ | - | - | - | - | 0.162 | 0.423 | 0.239 | 0.204 |
| $k_v = 27$ | - | - | - | - | 0.160 | 0.422 | 0.243 | 0.204 |

**Effect of variate Reordering.** Our model's reliance on local convolutions suggests that the ordering of variates could influence performance. To investigate this, we conduct a rigorous experiment across all eight benchmark datasets. We fix the model configuration to have a small cross-variate kernel size ($k_v = 3$) and a stack of two VarTCNBlocks, resulting in an effective receptive field of 5 across the variate axis. This constrained setup is designed to be highly sensitive to the local variate neighborhood. We compare the model's performance under four distinct ordering strategies: Original Ordering, Random Ordering, Degree Ordering, and Spectral Ordering (see Appendix I for details).

The results, summarized in Table 3, are counter-intuitive yet illuminating. The model exhibits a surprising degree of robustness to the variate ordering, as all four strategies yield remarkably similar performance. This finding suggests that correlation-based sorting methods, which operate by grouping highly similar variates, may not provide the most effective dependency signals. The dependencies captured by VPNet appear to be more complex than simple instantaneous correlations, implying that other factors, such as time-lagged relationships, may play a more critical role.

Table 3: Ablation study on variate reordering strategies. We report the average MSE across four prediction lengths 96, 192, 336, 720 for each of the eight benchmarks, with a fixed input length of 96. All models use a fixed configuration ($k_v = 3, N = 2$ layers). Lower is better.

| VPNET | ETTh1 | ETTh2 | ETTm1 | ETTm2 | ECL | Traffic | Weather | Solar |
|---|---|---|---|---|---|---|---|---|
| Original | 0.435 | 0.362 | 0.378 | 0.275 | 0.171 | 0.431 | 0.247 | 0.205 |
| Spectral | 0.436 | 0.366 | 0.374 | 0.275 | 0.174 | 0.428 | 0.246 | 0.209 |
| Degree | 0.436 | 0.365 | 0.378 | 0.276 | 0.173 | 0.433 | 0.247 | 0.209 |
| Random | 0.439 | 0.362 | 0.378 | 0.275 | 0.171 | 0.431 | 0.247 | 0.205 |

## 4.4 LOCAL MODELING MECHANISMS ANALYSIS

To further investigate the efficacy of local cross-variate modeling, we implemented two Transformer-based variants: **LANet (Local-Window Attention)** and **SANet (Sparse Attention)**. The quantitative results across eight benchmark datasets are summarized in **Table 4**. As shown in the table, while VP-Net generally outperforms the attention-based variants (particularly on dense datasets like Electricity and Traffic), the variants achieve competitive performance on datasets with different characteristics, prompting a deeper analysis of the trade-offs between TCN-based and Attention-based approaches.

**Structural Prior vs. Content-Based Addressing.** The fundamental distinction lies in the dependency capture mechanism. Transformer-based variants rely on *content-based addressing*, dynamically computing attention weights (Softmax($QK^T$)) to learn instance-specific relationships. Theoretically, this offers a higher representational ceiling by capturing arbitrary dependencies without geometric constraints. Conversely, VPNet (TCN-based) enforces a strong *inductive bias* through fixed convolutional kernels, treating the multivariate input as a topological grid. While this "static" modeling appears less flexible, our results suggest that this strong structural prior acts as an effective regularizer. It leads to superior optimization stability and generalization on dense datasets, whereas fully data-driven attention mechanisms often struggle with optimization or require larger data regimes to converge effectively.

**Robustness and Variate Ordering.** In the **Dense-Dependency Scenario** (e.g., datasets such as Traffic and Electricity with dense correlations and high information redundancy), VPNet exhibits pronounced robustness to variate ordering. This is because even under random permutations of the variables, each local neighborhood can still cover multiple subsets of correlated variates. As a result, in such settings, VPNet serves as a robust and efficient modeling strategy.

In contrast, in the **Sparse-Dependency Scenario** (e.g., Hetero-Mix or datasets with sparse dependency structures), variate ordering has a much more significant impact on VPNet's performance. In these cases, if the correlation structure among variates is ignored and the ordering is randomly permuted, a single local receptive field is likely to contain many irrelevant or weakly correlated variates, which reduces the concentration of informative signals. By adopting **correlation-aware or structure-aware ordering strategies**, one can "compress" more useful dependencies into each receptive field, substantially improving the local signal-to-noise ratio and allowing the convolutional kernels to enhance VPNet's performance *without increasing the number of parameters*.

Table 4: Long-term forecasting performance comparison for horizons of $\{96, 192, 336, 720\}$. We compare three variants: **VPNet** (Raw), **LANet** (Local), and **SANet** (Sparse). The input sequence length is fixed to 96. The best results are highlighted in **bold**.

| DataSets | | Weather | | Solar-Energy | | Electricity | | Traffic | | ETTh1 | | ETTh2 | | ETTm1 | | ETTm2 | |
|---|---|---|---|---|---|---|---|---|---|---|---|---|---|---|---|---|---|
| Metric | | MSE | MAE | MSE | MAE | MSE | MAE | MSE | MAE | MSE | MAE | MSE | MAE | MSE | MAE | MSE | MAE |
| VPNet | 96 | **0.157** | **0.192** | **0.177** | **0.178** | **0.135** | **0.224** | **0.384** | 0.258 | **0.374** | **0.386** | **0.284** | 0.330 | **0.313** | 0.342 | **0.169** | **0.244** |
| | 192 | **0.207** | **0.239** | 0.196 | **0.206** | **0.151** | **0.239** | **0.406** | 0.266 | **0.428** | **0.418** | 0.357 | **0.376** | **0.362** | **0.371** | **0.233** | **0.288** |
| | 336 | **0.258** | **0.281** | **0.216** | **0.219** | **0.166** | **0.257** | **0.429** | 0.275 | **0.464** | **0.439** | **0.392** | **0.406** | **0.385** | **0.389** | **0.291** | **0.327** |
| | 720 | **0.330** | **0.333** | 0.228 | **0.225** | **0.196** | **0.284** | 0.466 | 0.294 | **0.471** | **0.465** | 0.391 | **0.418** | **0.444** | **0.424** | **0.387** | **0.387** |
| | Avg | **0.238** | **0.261** | **0.204** | **0.207** | **0.162** | **0.251** | **0.421** | 0.273 | **0.434** | **0.427** | 0.356 | **0.383** | **0.376** | **0.382** | **0.270** | **0.312** |
| LANet | 96 | **0.157** | 0.193 | 0.185 | 0.196 | 0.144 | 0.230 | 0.392 | **0.257** | 0.377 | 0.388 | 0.288 | 0.330 | **0.313** | **0.340** | 0.177 | 0.249 |
| | 192 | 0.211 | 0.245 | **0.194** | 0.207 | 0.158 | 0.243 | 0.413 | **0.265** | 0.430 | 0.420 | 0.356 | 0.377 | 0.369 | 0.372 | 0.243 | 0.294 |
| | 336 | 0.277 | 0.293 | **0.216** | 0.222 | 0.173 | 0.259 | **0.429** | **0.272** | 0.468 | 0.441 | 0.393 | 0.407 | 0.396 | 0.394 | 0.300 | 0.333 |
| | 720 | 0.353 | 0.342 | **0.225** | 0.229 | 0.208 | 0.291 | **0.462** | **0.293** | 0.485 | 0.468 | 0.415 | 0.431 | 0.456 | 0.428 | 0.393 | 0.389 |
| | Avg | 0.250 | 0.268 | 0.205 | 0.214 | 0.171 | 0.256 | 0.424 | **0.272** | 0.440 | 0.429 | 0.363 | 0.386 | 0.384 | 0.384 | 0.278 | 0.316 |
| SANet | 96 | 0.158 | 0.193 | 0.183 | 0.197 | 0.149 | 0.233 | 0.394 | 0.259 | 0.376 | 0.388 | **0.284** | **0.329** | **0.313** | **0.340** | 0.175 | 0.248 |
| | 192 | 0.213 | 0.246 | **0.194** | **0.206** | 0.162 | 0.245 | 0.413 | 0.267 | 0.430 | 0.420 | **0.354** | 0.377 | 0.367 | **0.371** | 0.241 | 0.293 |
| | 336 | 0.277 | 0.291 | 0.217 | 0.222 | 0.175 | 0.259 | **0.429** | 0.274 | 0.468 | 0.442 | **0.392** | 0.407 | 0.398 | 0.393 | 0.301 | 0.334 |
| | 720 | 0.361 | 0.345 | 0.227 | 0.230 | 0.207 | 0.288 | 0.465 | 0.295 | 0.486 | 0.473 | 0.412 | 0.430 | 0.457 | 0.428 | 0.392 | 0.388 |
| | Avg | 0.252 | 0.269 | 0.205 | 0.214 | 0.173 | 0.256 | 0.425 | 0.274 | 0.440 | 0.431 | 0.361 | 0.386 | 0.384 | 0.383 | 0.277 | 0.316 |

## 4.5 Model Efficiency Analysis

For practical deployment in high-dimensional forecasting, computational and memory efficiency are as critical as predictive accuracy. We therefore evaluate efficiency on the two datasets with the largest number of variates, **Electricity** ($C = 321$) and **Traffic** ($C = 862$), comparing VPNet with leading baselines including iTransformer, PatchTST, TimeMixer, Crossformer, and Pathformer. Our analysis considers the joint trade-off among accuracy (MSE), training time per batch, and GPU memory usage.

Figure 4 summarizes the results, revealing a clear accuracy–efficiency frontier. VPNet consistently achieves the most favorable balance, delivering state-of-the-art accuracy at competitive computational cost. On both datasets, VPNet attains the lowest MSE, demonstrating the effectiveness of its local cross-variate modeling.

A direct comparison with **iTransformer** illustrates the difference in scaling behavior. iTransformer achieves the fastest training times due to its simple design, but its reliance on variate-wise attention induces quadratic memory scaling. As the number of variates grows from 321 to 862, its peak memory nearly doubles ($+99\%$, from 2174MB to 4376MB). In contrast, VPNet's memory footprint increases by only $67\%$ (from 3308MB to 5520MB), consistent with its linear complexity in the variate dimension.

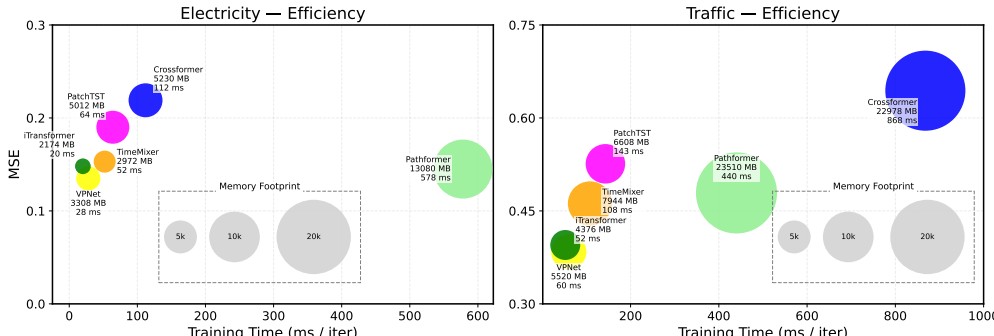

Figure 4: Efficiency comparison on **Electricity** ($C = 321$) and **Traffic** ($C = 862$). We report mean squared error (MSE), training time per batch, and peak GPU memory usage for VPNet and baseline models.

At the opposite extreme, global dependency models such as Crossformer and Pathformer incur prohibitive costs in both computation and memory, rendering them impractical for large-scale use. Conversely, TimeMixer achieves strong efficiency but sacrifices accuracy due to its channel-independent design, underperforming on complex, high-dimensional datasets.

VPNet establishes itself at the accuracy–efficiency Pareto frontier. By grounding its design in The Local Sufficiency Hypothesis, it delivers both superior forecasting accuracy and scalable efficiency, making it a practical solution for real-world, large-scale forecasting tasks.

## 5 RELATED WORK

**Channel Independence.** Channel independence has emerged as a simple yet effective strategy for multivariate time series forecasting. The core idea is to model each variate independently, thereby avoiding the "negative transfer" that may arise from noisy or spurious cross-variate correlations. PatchTST (Nie et al., 2023) exemplifies this paradigm by combining a patching strategy with a channel-independent Transformer architecture. Similarly, lightweight models such as TimeMixer and DLinear (Zeng et al., 2023) have shown that accurate univariate forecasting can achieve strong performance with high computational efficiency. However, these methods often underperform in high-dimensional settings where cross-variate dependencies are critical.

**Channel Fusion.** In contrast, channel fusion aims to explicitly capture dependencies across variates (Liu et al., 2025a). Representative approaches include Crossformer (Zhang & Yan, 2023), Pathformer (Chen et al., 2024), iTransformer (Liu et al., 2024a), and CARD (Wang et al., 2024b). While such methods generally outperform channel-independent models on high-dimensional datasets, their computational and memory costs grow rapidly with the number of variates, limiting scalability. To address this challenge, we propose the local sufficiency hypothesis. By focusing on capturing local cross-variate dependencies, our proposed method reconciles the trade-off between modeling expressiveness and computational cost. It achieves new state-of-the-art (SOTA) performance while ensuring that computational complexity scales linearly with the number of variates, making it a practical and effective solution for high-dimensional forecasting tasks.

## 6 CONCLUSION

In this work, we revisit the long-standing assumption that global dependency modeling is indispensable for high-dimensional time series forecasting. We formalize the *Local Sufficiency Hypothesis*, which posits that local cross-variate interactions are often sufficient to retain predictive power while avoiding the inefficiencies of global mixing. Building on this principle, we introduce **VPNet**, a new architecture that leverages a variate–patch field representation and the VarTCNBlock to model local dependencies with linear scalability. Through comprehensive evaluation on eight public benchmarks, VPNet achieves new state-of-the-art results. These findings demonstrate that focusing on local sufficiency provides a principled and scalable solution to the critical accuracy–efficiency trade-off in multivariate forecasting.

ACKNOWLEDGMENTS

This work was supported by the Guizhou Provincial Program on Commercialization of Scientific and Technological Achievements (Qiankehezhongyindi [2025] No. 006) and Seek Data Group, Emoney Inc.

ETHICS STATEMENT

This research is based on publicly available, anonymized datasets commonly used for benchmarking in the time series forecasting community. The work focuses on foundational modeling techniques for general forecasting tasks, such as predicting electricity consumption and traffic patterns. We do not foresee any direct negative societal impacts or ethical concerns arising from our methodology or its applications. Our research adheres to the principles of ethical academic conduct.

REPRODUCIBILITY STATEMENT

To ensure the full reproducibility of our results, we provide the following resources.

**Code**    The complete source code for VPNet, along with scripts to run all experiments reported in this paper, is provided in the Supplementary Materials. The code is also available at this repository: https://github.com/iuaku/VPNet/.

**Data**    All eight datasets used in our evaluation (Weather, Traffic, Electricity, Solar-Energy, and ETT benchmarks) are publicly available and can be downloaded from the dataset links provided in the official TimeMixer (Wang et al., 2024a) source code.

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

## A  LLM Usage Statement

In the preparation of this manuscript, we utilized Large Language Models (LLMs) as writing assistants. The use of these tools was strictly limited to improving the clarity, grammar, and overall style of the text. No part of the core scientific contributions, including the formulation of the hypothesis, the design of the model architecture, the implementation of the code, the generation of experimental results, or the analysis of those results, was produced by LLMs. All suggestions from these tools were critically reviewed, edited, and manually incorporated by the authors to ensure the final text accurately reflects our own work and ideas.

## B  Implementation Details

**Dataset Details.**  This section provides a detailed description of the public benchmark datasets used for the empirical evaluation of our model in multivariate time series forecasting. For data preprocessing, splitting, and normalization, we adhere to the standard protocols established in widely-recognized previous works (Zhou et al., 2021; Wu et al., 2021). A summary of the key statistical properties of each dataset is presented in Table 5. The evaluation suite includes several standard benchmarks. The ETT (Electricity Transformer Temperature) collection contains data from two electricity transformers with 7 variates, recorded at hourly (ETTh1, ETTh2) and 15-minute intervals (ETTm1, ETTm2) from 2016 to 2018. The Electricity (ECL) dataset contains the hourly power consumption of 321 clients from 2016 to 2019. The Weather dataset comprises 21 meteorological indicators from Germany, collected every 10 minutes during 2020. The Traffic dataset documents hourly road occupancy rates from 862 sensors in the San Francisco Bay Area from 2015 to 2016. Finally, the Solar-Energy dataset records solar power generation from 137 photovoltaic (PV) plants at 10-minute intervals during 2006.

**Experiment Details.**  All experiments were implemented in PyTorch and conducted on a single NVIDIA A100 40GB GPU. For model optimization, we employ the Adam optimizer (Kingma & Ba, 2017) with an initial learning rate of $1 \times 10^{-4}$ and a batch size of 32. To remain consistent with prior works, we use a fixed look-back window of $L = 96$. For the core model hyperparameters, we select the number of VarTCNBlocks $N$ from $\{1, 2, 3\}$, the patch length $p$ from $\{8, 16\}$, and the

hidden dimension $H$ from $\{64, 128, 256\}$ based on validation set performance for each dataset. Mean Squared Error (MSE) and Mean Absolute Error (MAE) are used as the primary evaluation metrics. For baselines where the experimental settings align with our main study, we directly report the results from TimeMixer (Wang et al., 2024a). In other cases, we reproduced the baseline results using the benchmark framework from the Time-Series Library [1].

Table 5: Statistics of the benchmark datasets.

| Dataset | Dim | Series Length | Dataset Size | Frequency | Information |
|---------|-----|---------------|--------------|-----------|-------------|
| ETTh1 | 7 | {96, 192, 336, 720} | (34465, 11521, 11521) | Hourly | Temperature |
| ETTh2 | 7 | {96, 192, 336, 720} | (34465, 11521, 11521) | Hourly | Temperature |
| ETTm1 | 7 | {96, 192, 336, 720} | (8545, 2881, 2881) | 15 min | Temperature |
| ETTm2 | 7 | {96, 192, 336, 720} | (8545, 2881, 2881) | 15 min | Temperature |
| Electricity | 321 | {96, 192, 336, 720} | (18317, 2633, 5261) | Hourly | Electricity |
| Weather | 21 | {96, 192, 336, 720} | (36792, 5271, 10540) | 10 min | Weather |
| Traffic | 862 | {96, 192, 336, 720} | (12185, 1757, 3509) | Hourly | Transportation |
| Solar-Energy | 137 | {96, 192, 336, 720} | (36601, 5161, 10417) | 10 min | Electricity |

## C    PROOFS AND PROBABILISTIC DESIGN RULES

We formalize the probabilistic statement used in the introduction and provide proofs.

Table 6: Probability Analysis for Different Parameters

| Info. Variate Ratio | Probability for $\mathcal{E}_k$ | | | |
|---------------------|---------|---------|---------|---------|
| (r/C) | k=1 | k=3 | k=7 | k=17 |
| 10% | 0.095 | 0.259 | 0.799 | 0.999 |
| 30% | 0.259 | 0.593 | 0.996 | 1.000 |
| 50% | 0.393 | 0.777 | 1.000 | 1.000 |
| 70% | 0.503 | 0.878 | 1.000 | 1.000 |
| 90% | 0.593 | 0.931 | 1.000 | 1.000 |

**Theorem C.1** (Hypergeometric exact probability for zero hits). *Let $C \geq 2$ be the total number of variates and suppose that for a fixed target variate there are exactly $r$ "informative" variates among the other $C - 1$ variates (i.e., $0 \leq r \leq C - 1$). Consider selecting a subset of $k$ distinct variates uniformly at random from the $C - 1$ non-target variates (this models the set of variates falling into a fixed contiguous window under a uniformly random permutation). Let $X$ be the number of informative variates in the chosen subset. Then*

$$\mathscr{P}r[X = 0] \;=\; \frac{\binom{C-1-r}{k}}{\binom{C-1}{k}}. \tag{18}$$

---

[1] https://github.com/thuml/Time-Series-Library

Table 7: Confidence Intervals for Different Parameters

| Info. Variate Ratio | k=7 | | k=17 | |
|---------------------|-----|-----|------|-----|
| (r/C) | 95% | 99% | 95% | 99% |
| 10% | [0, 2] | [0, 3] | [0, 4] | [0, 5] |
| 30% | [0, 5] | [0, 6] | [1, 9] | [0, 10] |
| 50% | [1, 6] | [0, 7] | [3, 14] | [2, 15] |
| 70% | [2, 7] | [1, 7] | [7, 16] | [6, 17] |
| 90% | [5, 7] | [4, 7] | [12, 17] | [12, 17] |

*Proof.* There are $\binom{C-1}{k}$ equally likely ways to choose a $k$-subset from the $C-1$ non-target variates. The number of choices that contain zero informative variates is the number of ways to choose all $k$ elements from the $C-1-r$ non-informative variates, which is $\binom{C-1-r}{k}$. Dividing the favorable count by the total count yields equation 18. $\square$

**Corollary C.1** (Exponential upper bound). *Under the same notation as Theorem C.1, the zero-hit probability satisfies*

$$\mathscr{P}r[X=0] \;\leq\; \left(1 - \frac{r}{C-1}\right)^{k} \;\leq\; \exp\left(-\frac{kr}{C-1}\right). \tag{19}$$

*Proof.* Starting from equation 18 we write the ratio form

$$\mathscr{P}r[X=0] \;=\; \mathscr{P}rod_{i=0}^{k-1} \frac{C-1-r-i}{C-1-i}.$$

For each factor we have

$$\frac{C-1-r-i}{C-1-i} \;=\; 1 - \frac{r}{C-1-i} \;\leq\; 1 - \frac{r}{C-1},$$

because the denominator $C-1-i$ is at most $C-1$ for $i \geq 0$. Taking the product yields the first inequality in equation 19. The second inequality follows from $(1-x) \leq e^{-x}$ applied to $x = \frac{r}{C-1}$ and exponentiation to the power $k$. $\square$

**Corollary C.2** (Design rule for at-least-one hit). *If we require that a randomly chosen window of width $k$ contains at least one informative variate with probability at least $1-\delta$, i.e.*

$$\mathscr{P}r[X \geq 1] \geq 1 - \delta,$$

*then it suffices to choose $k$ satisfying*

$$k \;\geq\; \frac{C-1}{r} \ln\frac{1}{\delta}. \tag{20}$$

*Proof.* From Corollary C.1 we have

$$\mathscr{P}r[X \geq 1] = 1 - \mathscr{P}r[X=0] \geq 1 - \exp\left(-\frac{kr}{C-1}\right).$$

Requiring $1 - \exp(-kr/(C-1)) \geq 1-\delta$ is equivalent to $\exp(-kr/(C-1)) \leq \delta$, which rearranges to equation 20. $\square$

**Concentration around the mean.** Let $\mu = \mathbb{E}[X] = k \cdot \frac{r}{C-1}$ denote the hypergeometric mean. Standard concentration bounds for the hypergeometric distribution (which can be derived from Hoeffding's inequality or by coupling to an appropriate binomial distribution) give that for any $0 < \varepsilon < 1$,

$$\mathscr{P}r\left[X \leq (1-\varepsilon)\mu\right] \;\leq\; \exp\left(-\frac{\varepsilon^2 \mu}{2}\right). \tag{21}$$

A corresponding upper tail bound holds:

$$\mathscr{P}r\left[X \geq (1+\varepsilon)\mu\right] \;\leq\; \exp\left(-\frac{\varepsilon^2 \mu}{3}\right).$$

These inequalities quantify that once $\mu$ is moderate, the number of informative variates inside a random window concentrates tightly around $\mu$.

## D  EMPIRICAL ANALYSIS OF DATASET CHARACTERISTICS

To empirically validate the *Local Sufficiency Hypothesis*, we analyzed the intrinsic properties of all benchmark datasets. We focus on three key aspects: (1) **Granger Dependency Density**, measured by the ratio of significant Granger Causal edges ($p < 0.01$, lag=3) after differencing and global mean removal; (2) **Information Redundancy**, quantified by Principal Component Analysis (PC1 variance and Effective Dimension Compression Ratio); and (3) **Correlation Strength**, measured by the distribution of pairwise Pearson correlation coefficients. The quantitative results are summarized in Table 8:

Table 8: Statistical analysis of dataset characteristics. **Granger Density** indicates the ratio of significant causal pairs. **PC1 Var** denotes the variance explained by the first principal component. **Compression Ratio** is the total number of variates divided by the number of components required to explain 95% variance. **Correlation** metrics show the percentage of variate pairs exceeding absolute correlation thresholds. **Bold** highlights the highest values indicating extreme density.

| Dataset | Granger | PCA (Redundancy) | | | variate Correlation (Pearson $|r|$) | | | |
|---|---|---|---|---|---|---|---|---|
| | Density | PC1 Var | 95% Comps | Comp. Ratio | Avg $|r|$ | > 0.3 | > 0.5 | > 0.7 |
| **Solar** | 51.87% | **91.77%** | **4** / 136 | **34.00x** | **0.9167** | **100.0%** | **100.0%** | **100.0%** |
| **Traffic** | 69.45% | 57.67% | 202 / 862 | 4.27x | 0.5638 | 89.92% | 66.84% | 25.67% |
| **Electricity** | **89.19%** | 54.68% | 86 / 321 | 3.73x | 0.4893 | 68.29% | 46.37% | 32.49% |
| **Weather** | 64.52% | 42.44% | 9 / 21 | 2.33x | 0.2956 | 35.71% | 25.24% | 20.95% |
| **ETTh1** | 64.29% | 34.39% | 5 / 7 | 1.40x | 0.2221 | 19.05% | 9.52% | 9.52% |
| **ETTh2** | 52.38% | 43.10% | 5 / 7 | 1.40x | 0.3246 | 42.86% | 28.57% | 4.76% |
| **ETTm1** | 47.62% | 34.57% | 5 / 7 | 1.40x | 0.2243 | 19.05% | 9.52% | 9.52% |
| **ETTm2** | 30.95% | 43.08% | 5 / 7 | 1.40x | 0.3245 | 42.86% | 28.57% | 4.76% |

**Analysis.**

- **Large-scale datasets with high redundancy.** The Solar, Traffic, and Electricity datasets exhibit distinct characteristics of *dense regimes*. They possess high effective dimension compression ratios ($> 3.7\times$) and strong variate correlations. Notably, Electricity maintains an exceptionally high causal density (89.19%) even after removing global trends, suggesting a ubiquitous local interaction network. This justifies the superior performance of VPNet on these datasets, as local kernels can efficiently aggregate the redundant and dense information.
- **Lower-dimensional datasets with weaker cross-variate dependencies.** Conversely, the ETT datasets show significantly lower redundancy (Compression Ratio $\approx 1.4\times$) and weaker correlations (strong correlations $|r| > 0.5$ are generally $< 10\%$).

# E    EVALUATION ON VARIATE ORDERING USING A HETEROGENEOUS COMPOSITE DATASET

To examine how variate ordering affects local modeling in datasets with heterogeneous dependency patterns, we constructed a composite dataset named **Hetero-Mix** by concatenating the variates from three benchmark datasets—Traffic, Electricity, and Weather—along the channel dimension. Unlike homogeneous datasets where dense correlations reduce sensitivity to ordering, Hetero-Mix brings together variables with distinct statistical characteristics, making it suitable for testing how ordering influences local receptive field models.

We evaluated VPNet and a local-attention variant (LANet) under two ordering conditions: (1) **Clustered**, where variates are grouped according to their dataset of origin; and (2) **Shuffled**, where all variates are randomly permuted. As shown in Table 9, both models achieve better accuracy under the Clustered setting. This demonstrates that when correlations vary substantially across groups of variables, aligning the input ordering with the underlying dependency patterns improves the effectiveness of localized modeling.

# F    STRESS TEST UNDER EXTREME NOISE AND SPARSE DEPENDENCIES

To rigorously evaluate model robustness in scenarios dominated by sparse or long-range cross-variate dependencies, we conducted a stress test using the Electricity dataset injected with **$10\times$ Gaussian noise** ($10 \times \sigma_{\text{noise}}$). This setup creates an extremely low signal-to-noise ratio (SNR) regime, effectively disintegrating dense local correlations. To compensate for the high noise level and capture dispersed signals, we adjusted the VPNet configuration to use a larger receptive field (Kernel Size=27, Layers=2).

We tested VPNet using two variable orderings: (1) **Clustered**, where variates follow their original dataset indexing; and (2) **Shuffled**, where the ordering is randomized. Table 10 shows that VPNet

Table 9: Performance comparison on the **Hetero-Mix** dataset. "Clustered" preserves the original grouping [Traffic ⊕ Electricity ⊕ Weather], while "Shuffled" randomly permutes variates. The results show that both models benefit from topology-aware ordering (Clustered).

| Ordering Setting | Horizon | VPNet | | LANet | |
|---|---|---|---|---|---|
| | | MSE | MAE | MSE | MAE |
| **Clustered** (Ordered) | 96 | **0.345** | **0.264** | **0.347** | **0.264** |
| | 192 | **0.368** | **0.274** | **0.369** | **0.274** |
| | 336 | **0.390** | **0.286** | **0.387** | **0.284** |
| | 720 | **0.435** | **0.313** | **0.432** | **0.314** |
| | *Avg* | **0.385** | **0.284** | **0.384** | **0.284** |
| **Shuffled** (Random) | 96 | 0.349 | 0.270 | 0.352 | 0.268 |
| | 192 | 0.369 | 0.278 | 0.371 | 0.275 |
| | 336 | 0.393 | 0.288 | 0.389 | 0.286 |
| | 720 | 0.440 | 0.317 | 0.433 | 0.314 |
| | *Avg* | 0.388 | 0.288 | 0.386 | 0.286 |

performs consistently better with the original ordering. This indicates that when the signal is weak, variable orderings that place moderately correlated variates closer together help local models extract meaningful structure more effectively.

Additionally, the large-kernel configuration allows VPNet to approach the behavior of broader-receptive-field models, suggesting that increasing the receptive field is a practical adaptation strategy when dependency patterns become weak or diffuse.

Table 10: Stress test on Electricity + $10\times$ Noise. The model configuration is fixed at [Layers=2, Kernel=27]. "Ordered" denotes structural prior-based clustering, while "Shuffled" denotes random permutation. **Bold** indicates the best performance.

| Horizon | Electricity$_{\text{Noise}}$(Ordered) | | Electricity$_{\text{Noise}}$(Shuffled) | |
|---|---|---|---|---|
| | MSE | MAE | MSE | MAE |
| 96 | **0.137** | **0.232** | 0.148 | 0.239 |
| 192 | **0.153** | **0.248** | 0.161 | 0.252 |
| 336 | **0.174** | **0.270** | 0.178 | 0.270 |
| 720 | **0.199** | **0.295** | 0.224 | 0.312 |
| *Avg* | **0.166** | **0.261** | 0.178 | 0.268 |

## G  VISUALIZATION OF LEARNED DEPENDENCIES AND RECEPTIVE FIELDS

To better illustrate the dependencies learned by VPNet and verify its capability to capture local cross-variate interactions, we conducted a **gradient-based saliency analysis**. We computed the absolute gradients of the prediction output with respect to the input ($|\partial\hat{y}/\partial\mathbf{X}|$), averaged over 64 samples for stability.

**Methodology.** We present three complementary visualizations: (1) a **saliency heatmap** showing the spatio-temporal receptive field, (2) **global variate importance** summarizing cross-variate contributions, and (3) **temporal importance** reflecting how historical information is utilized.

**Analysis.** The visualizations reveal dataset-specific dependency patterns, summarized as follows:

- **ETTh1: Predominantly Auto-Regressive Behavior.** As shown in Figure 5, when predicting variate 2 at the first step, gradients concentrate almost entirely on the variate's own history. Neighboring variates contribute minimally. This aligns with our earlier statistical findings indicating weak cross-variate correlations in ETTh1. The temporal importance plot also shows periodic spikes, demonstrating that VPNet captures seasonal patterns even when cross-variate signals are limited.

- **Electricity: Discovery of Dataset-Specific Couplings.** In Figure 6, when predicting variate 200, the model assigns substantial importance to the history of variate 182, rather than relying only on auto-regression. This indicates that VPNet can detect meaningful cross-variate relationships within its receptive field and exploit them for prediction.
- **Traffic: Utilization of Local Neighborhood Clusters.** In the Traffic dataset (Figure 7), the prediction of variate 400 activates a wide band of nearby variates. This reflects a collective neighborhood influence, where VPNet aggregates information from a group of highly correlated variates to infer near-future behavior, which is consistent with the strongly correlated structure observed in Traffic.

In summary, these visualizations demonstrate that VPNet adapts naturally to the dependency characteristics of each dataset: it behaves auto-regressively when cross-variate signals are weak (ETTh1), identifies meaningful pairwise couplings when present (Electricity), and aggregates rich local neighborhoods when the data exhibit strong spatial correlations (Traffic).

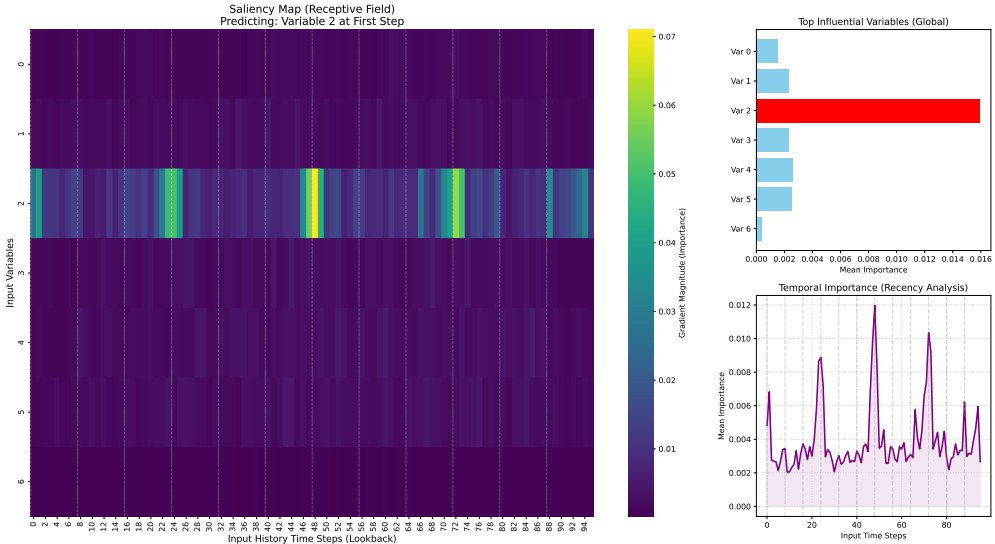

Figure 5: **Interpretability Analysis on ETTh1.** Visualization for predicting variate 2 at the first step. The saliency heatmap and variate importance indicate that the model primarily relies on the variate's own temporal history, with minimal cross-variate contribution. This is consistent with the weak inter-variate correlations observed in ETTh1. The temporal importance curve also exhibits clear periodicity.

# H ADDITIONAL EVALUATION ON GIFT-EVAL BENCHMARK

## H.1 EXPERIMENTAL SETUP AND IMPLEMENTATION

We acknowledge the recent emergence of comprehensive benchmarks aimed at standardizing the evaluation of forecasting models (Liu et al., 2024b; Fan et al., 2025), such as the **FEV-benchmark** (Shchur et al., 2025) and **GIFT-Eval** (Aksu et al., 2024). To further assess the generalization capability of VPNet in a unified evaluation environment, we extended our experiments to the **GIFT-Eval** benchmark.

To ensure compatibility with the benchmark protocol, we integrated VPNet into the `gluonts` framework and added a probabilistic projection head to support both point and probabilistic forecasting.

## H.2 PERFORMANCE COMPARISON

We compared VPNet against ITransformer, a state-of-the-art baseline under the GIFT-Eval setup. The evaluation metric is the Mean Absolute Percentage Error (MAPE).

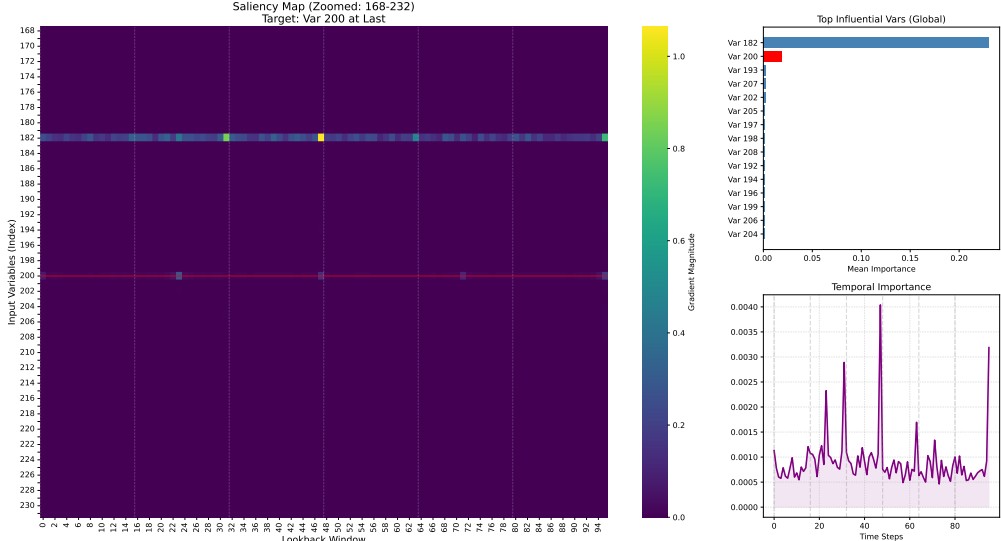

Figure 6: **Interpretability Analysis on Electricity.** Visualization for predicting variate 200. The global variate importance plot shows that **variate 182** contributes more significantly than the target variate itself, demonstrating that VPNet successfully identifies strong and meaningful cross-variate dependencies within its receptive field.

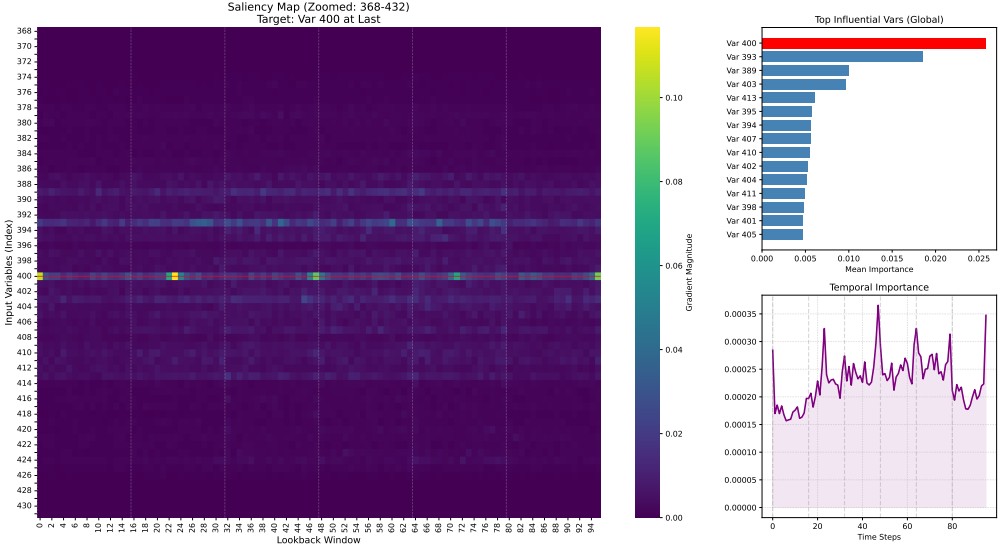

Figure 7: **Interpretability Analysis on Traffic.** Visualization for predicting variate 400. The heatmap exhibits strong activations across a wide group of highly correlated variates, indicating that the prediction relies on the collective information from multiple related variables rather than any single one. This highlights VPNet's ability to effectively utilize locally correlated patterns in datasets with rich cross-variate structure.

The results presented in Table 11 show that VPNet achieves competitive performance under a standardized evaluation pipeline, outperforming ITransformer on **17 out of 33** datasets/settings. These findings indicate that VPNet maintains strong robustness and generalization capability across diverse forecasting tasks, further supporting the effectiveness of its locality-driven modeling strategy.

Table 11: Performance comparison on GIFT-Eval benchmark (Metric: MAPE). Best results are highlighted in **bold**. VPNet achieves superior performance in 17 settings, demonstrating robustness even under univariate input conditions.

| Dataset Setting | VPNet (Ours) | ITransformer | Winner |
|---|---|---|---|
| bitbrains_fast_storage/5T/short | 2.178 | **1.689** | ITransformer |
| bitbrains_fast_storage/H/short | 3.272 | **2.962** | ITransformer |
| bitbrains_rnd/5T/short | 1.522 | **1.268** | ITransformer |
| bitbrains_rnd/H/short | 3.280 | **2.535** | ITransformer |
| bizitobs_application/10S/long | **0.054** | 0.065 | VPNet |
| bizitobs_application/10S/medium | **0.043** | 0.047 | VPNet |
| bizitobs_application/10S/short | **0.038** | 0.041 | VPNet |
| bizitobs_l2c/5T/long | **0.463** | 0.537 | VPNet |
| bizitobs_l2c/5T/medium | **0.461** | 0.567 | VPNet |
| bizitobs_l2c/5T/short | **0.148** | 0.174 | VPNet |
| bizitobs_l2c/H/long | **0.664** | 0.670 | VPNet |
| bizitobs_l2c/H/medium | 0.602 | **0.586** | ITransformer |
| bizitobs_l2c/H/short | **0.643** | 0.707 | VPNet |
| bizitobs_service/10S/long | 0.420 | **0.250** | ITransformer |
| bizitobs_service/10S/medium | 0.307 | **0.189** | ITransformer |
| bizitobs_service/10S/short | **0.147** | 0.161 | VPNet |
| ett1/15T/long | 0.797 | **0.787** | ITransformer |
| ett1/15T/medium | **0.735** | 0.749 | VPNet |
| ett1/15T/short | 0.641 | **0.622** | ITransformer |
| ett1/D/short | 1.697 | **1.623** | ITransformer |
| ett1/H/short | 0.511 | **0.506** | ITransformer |
| ett1/W/short | 0.882 | **0.770** | ITransformer |
| ett2/15T/long | **0.150** | 0.156 | VPNet |
| ett2/15T/medium | **0.144** | 0.152 | VPNet |
| ett2/15T/short | **0.139** | 0.162 | VPNet |
| ett2/D/short | **0.368** | 0.464 | VPNet |
| ett2/H/long | 0.220 | **0.219** | ITransformer |
| ett2/H/medium | 0.207 | **0.199** | ITransformer |
| ett2/H/short | **0.138** | 0.139 | VPNet |
| ett2/W/short | **0.159** | 0.292 | VPNet |
| jena_weather/10T/short | 0.519 | **0.416** | ITransformer |
| jena_weather/D/short | **0.887** | 1.120 | VPNet |
| jena_weather/H/short | 1.427 | **1.320** | ITransformer |

## I    VARIATE ORDERING STRATEGIES

To investigate the impact of variate arrangement on locality-based architectures, we consider four ordering strategies. Each strategy reflects a distinct principle for structuring the variate dimension:

- **Original Ordering.** variates are preserved in the order provided by the dataset. This ordering reflects any implicit structure imposed during data collection (e.g., spatial layout of sensors or industry grouping of assets). It serves as a natural baseline.

- **Random Ordering.** variates are permuted uniformly at random. This destroys any pre-existing adjacency structure and thus provides a neutrality test. If a model still performs well under random ordering, it suggests robustness to locality assumptions.

- **Degree Ordering.** variates are ranked by their aggregate similarity (e.g., total correlation strength with others). The intuition is that highly connected variates are globally influential, and placing them adjacently increases the likelihood that local operators can capture their dependencies.

- **Spectral Ordering.** variates are arranged by the coordinates of the Fiedler vector of the graph Laplacian built from pairwise similarities. This spectral seriation seeks to embed

variates onto a line such that strongly related variates appear contiguously. It provides a principled way of linearizing high-dimensional dependency structures for local processing.

## J  EFFECT OF LOOK-BACK WINDOW SIZE

A model capable of capturing long-range temporal dependencies is expected to benefit from longer historical contexts (Zeng et al., 2022; Nie et al., 2023). To examine VPNet's ability to leverage historical information, we conduct an ablation study on the input sequence length $L$. We evaluate VPNet against two competitive baselines, iTransformer and TimeMixer, on the high-dimensional Electricity and Traffic datasets. The prediction horizon is fixed at $S = 96$, while the input length varies as $L \in \{48, 96, 144, 192, 240, 336\}$. Results are reported in Figure 8.

All three models generally improve as the look-back window increases, with lower MSE at larger $L$. However, their behaviors differ in how performance scales with context. iTransformer and TimeMixer exhibit gradual and consistent improvements across the full range of input lengths, indicating a steady reliance on longer histories. In contrast, VPNet reaches its best performance with substantially shorter contexts: its error decreases rapidly when $L$ increases from 48 to 144, after which further gains are marginal. On Electricity, VPNet achieves an MSE of 0.127 at $L = 144$, outperforming TimeMixer even at $L = 336$ (0.133). These results suggest that VPNet is able to extract the most relevant predictive patterns from moderate-length histories, highlighting its efficiency in utilizing contextual information without requiring excessively long sequences.

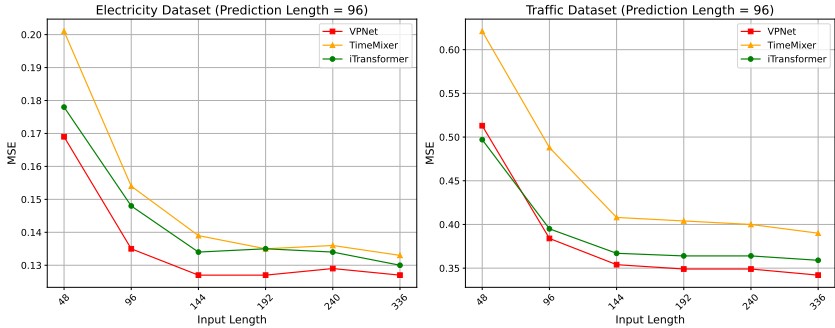

Figure 8: The effect of look-back window size on forecasting performance (MSE). We compare VPNet, iTransformer, and TimeMixer on the Electricity (left) and Traffic (right) datasets with a fixed prediction horizon of $S = 96$.

## K  FULL RESULTS

Due to the space limitation of the main text, we place the full results of all experiments in the following Table 12.

## L  SHOWCASES

For a qualitative assessment, we visualize the forecasts of a representative variate from the test set for each dataset (Figures 9, 10, 11, 12, 13, 14, 15, 16). These visualizations illustrate that VPNet's predictions consistently align more closely with the ground truth, adeptly capturing complex dynamics where other models falter.

## M  FUTURE WORK

Our findings open several promising avenues for future research. The surprising robustness of VPNet to variate ordering suggests that the model captures complex, dynamic relationships that

transcend static correlations. A deeper theoretical investigation into the nature of these time-lagged dependencies would be a valuable contribution. Another compelling direction is the development of architectures with adaptive locality. While our work shows the efficacy of a fixed local neighborhood, models that can learn to dynamically adjust the scope of their receptive field for different variates or layers could unlock further performance gains. Finally, the variate–patch field representation itself may prove to be a generalizable concept, and exploring its application to other multivariate sequence modeling tasks (e.g., spatio-temporal forecasting) is a promising direction for future research.

Table 12: Performance comparison on the long-term forecasting task for prediction horizons of $\{96, 192, 336, 720\}$ and their average. The input sequence length is fixed to 96. The best and second-best results on each dataset in each metric are highlighted in **bold red** and underlined blue fonts, respectively.

| Models | | VPNet (Ours) | | TimeKAN (2025) | | TimeMixer (2024a) | | iTransformer (2024a) | | Pathformer (2024) | | PatchTST (2023) | | Crossformer (2023) | | MICN (2023) | | TiDE (2023) | | TimesNet (2023a) | | DLinear (2023) | | FEDformer (2022) | |
|---|---|---|---|---|---|---|---|---|---|---|---|---|---|---|---|---|---|---|---|---|---|---|---|---|---|
| Metric | | MSE | MAE | MSE | MAE | MSE | MAE | MSE | MAE | MSE | MAE | MSE | MAE | MSE | MAE | MSE | MAE | MSE | MAE | MSE | MAE | MSE | MAE | MSE | MAE |
| Weather | 96 | 0.157 | 0.192 | 0.162 | 0.208 | 0.163 | 0.209 | 0.174 | 0.214 | 0.156 | 0.192 | 0.186 | 0.227 | 0.195 | 0.271 | 0.198 | 0.261 | 0.202 | 0.261 | 0.172 | 0.220 | 0.195 | 0.252 | 0.217 | 0.296 |
| | 192 | 0.207 | 0.239 | 0.207 | 0.249 | 0.208 | 0.250 | 0.221 | 0.254 | 0.206 | 0.240 | 0.234 | 0.265 | 0.209 | 0.277 | 0.239 | 0.299 | 0.242 | 0.298 | 0.219 | 0.261 | 0.237 | 0.295 | 0.276 | 0.336 |
| | 336 | 0.258 | 0.281 | 0.263 | 0.290 | 0.251 | 0.287 | 0.278 | 0.296 | 0.254 | 0.282 | 0.284 | 0.301 | 0.273 | 0.332 | 0.285 | 0.336 | 0.287 | 0.335 | 0.280 | 0.306 | 0.282 | 0.331 | 0.339 | 0.380 |
| | 720 | 0.330 | 0.333 | 0.338 | 0.340 | 0.339 | 0.341 | 0.358 | 0.347 | 0.340 | 0.336 | 0.356 | 0.349 | 0.379 | 0.401 | 0.351 | 0.388 | 0.351 | 0.386 | 0.365 | 0.359 | 0.345 | 0.382 | 0.403 | 0.428 |
| | Avg | 0.238 | 0.261 | 0.243 | 0.272 | 0.240 | 0.272 | 0.258 | 0.278 | 0.239 | 0.263 | 0.265 | 0.286 | 0.264 | 0.320 | 0.268 | 0.321 | 0.271 | 0.320 | 0.259 | 0.287 | 0.265 | 0.315 | 0.309 | 0.360 |
| Solar-Energy | 96 | 0.177 | 0.178 | 0.234 | 0.290 | 0.189 | 0.259 | 0.203 | 0.237 | 0.202 | 0.225 | 0.265 | 0.323 | 0.232 | 0.302 | 0.257 | 0.325 | 0.312 | 0.399 | 0.373 | 0.358 | 0.290 | 0.378 | 0.286 | 0.341 |
| | 192 | 0.196 | 0.206 | 0.274 | 0.309 | 0.222 | 0.283 | 0.233 | 0.261 | 0.235 | 0.245 | 0.288 | 0.332 | 0.371 | 0.410 | 0.278 | 0.354 | 0.339 | 0.416 | 0.397 | 0.376 | 0.320 | 0.398 | 0.291 | 0.337 |
| | 336 | 0.216 | 0.219 | 0.299 | 0.324 | 0.231 | 0.292 | 0.248 | 0.273 | 0.272 | 0.272 | 0.301 | 0.339 | 0.495 | 0.515 | 0.298 | 0.375 | 0.368 | 0.430 | 0.420 | 0.380 | 0.353 | 0.415 | 0.354 | 0.416 |
| | 720 | 0.228 | 0.225 | 0.295 | 0.318 | 0.223 | 0.285 | 0.249 | 0.275 | 0.255 | 0.256 | 0.295 | 0.336 | 0.526 | 0.542 | 0.299 | 0.379 | 0.370 | 0.425 | 0.420 | 0.381 | 0.357 | 0.413 | 0.380 | 0.437 |
| | Avg | 0.204 | 0.207 | 0.276 | 0.310 | 0.216 | 0.280 | 0.233 | 0.262 | 0.241 | 0.250 | 0.287 | 0.333 | 0.406 | 0.442 | 0.283 | 0.358 | 0.347 | 0.417 | 0.403 | 0.374 | 0.330 | 0.401 | 0.328 | 0.383 |
| Electricity | 96 | 0.135 | 0.224 | 0.174 | 0.266 | 0.153 | 0.247 | 0.148 | 0.240 | 0.145 | 0.236 | 0.190 | 0.296 | 0.219 | 0.314 | 0.180 | 0.293 | 0.237 | 0.329 | 0.168 | 0.272 | 0.210 | 0.302 | 0.193 | 0.308 |
| | 192 | 0.151 | 0.239 | 0.182 | 0.273 | 0.166 | 0.256 | 0.162 | 0.253 | 0.167 | 0.256 | 0.199 | 0.304 | 0.231 | 0.322 | 0.189 | 0.302 | 0.236 | 0.330 | 0.184 | 0.322 | 0.210 | 0.305 | 0.201 | 0.315 |
| | 336 | 0.166 | 0.257 | 0.197 | 0.286 | 0.185 | 0.277 | 0.178 | 0.269 | 0.186 | 0.275 | 0.217 | 0.319 | 0.246 | 0.337 | 0.198 | 0.312 | 0.249 | 0.344 | 0.198 | 0.300 | 0.223 | 0.319 | 0.214 | 0.329 |
| | 720 | 0.196 | 0.284 | 0.236 | 0.320 | 0.225 | 0.310 | 0.225 | 0.317 | 0.231 | 0.309 | 0.258 | 0.352 | 0.280 | 0.363 | 0.217 | 0.330 | 0.284 | 0.373 | 0.220 | 0.320 | 0.258 | 0.350 | 0.246 | 0.355 |
| | Avg | 0.162 | 0.251 | 0.197 | 0.286 | 0.182 | 0.273 | 0.178 | 0.270 | 0.182 | 0.269 | 0.216 | 0.318 | 0.244 | 0.334 | 0.196 | 0.309 | 0.252 | 0.344 | 0.193 | 0.304 | 0.225 | 0.319 | 0.214 | 0.327 |
| Traffic | 96 | 0.384 | 0.258 | 0.580 | 0.379 | 0.462 | 0.285 | 0.395 | 0.268 | 0.479 | 0.283 | 0.526 | 0.347 | 0.644 | 0.429 | 0.577 | 0.350 | 0.805 | 0.493 | 0.593 | 0.321 | 0.650 | 0.396 | 0.587 | 0.366 |
| | 192 | 0.406 | 0.266 | 0.550 | 0.363 | 0.473 | 0.296 | 0.417 | 0.276 | 0.484 | 0.292 | 0.522 | 0.332 | 0.665 | 0.431 | 0.589 | 0.356 | 0.756 | 0.474 | 0.617 | 0.336 | 0.598 | 0.370 | 0.604 | 0.373 |
| | 336 | 0.429 | 0.275 | 0.559 | 0.363 | 0.498 | 0.296 | 0.433 | 0.283 | 0.503 | 0.299 | 0.517 | 0.334 | 0.674 | 0.420 | 0.594 | 0.358 | 0.762 | 0.477 | 0.629 | 0.336 | 0.605 | 0.373 | 0.621 | 0.383 |
| | 720 | 0.466 | 0.294 | 0.600 | 0.381 | 0.506 | 0.313 | 0.467 | 0.302 | 0.537 | 0.322 | 0.552 | 0.352 | 0.683 | 0.424 | 0.613 | 0.361 | 0.719 | 0.449 | 0.640 | 0.350 | 0.645 | 0.394 | 0.626 | 0.382 |
| | Avg | 0.421 | 0.273 | 0.572 | 0.372 | 0.485 | 0.298 | 0.428 | 0.282 | 0.501 | 0.299 | 0.529 | 0.341 | 0.667 | 0.426 | 0.593 | 0.356 | 0.761 | 0.473 | 0.620 | 0.336 | 0.625 | 0.383 | 0.610 | 0.376 |
| ETTh1 | 96 | 0.374 | 0.386 | 0.374 | 0.397 | 0.375 | 0.400 | 0.386 | 0.405 | 0.390 | 0.390 | 0.460 | 0.447 | 0.423 | 0.448 | 0.426 | 0.446 | 0.479 | 0.464 | 0.384 | 0.402 | 0.397 | 0.412 | 0.395 | 0.424 |
| | 192 | 0.428 | 0.418 | 0.416 | 0.422 | 0.429 | 0.421 | 0.441 | 0.436 | 0.437 | 0.419 | 0.477 | 0.429 | 0.471 | 0.474 | 0.454 | 0.464 | 0.525 | 0.492 | 0.436 | 0.429 | 0.446 | 0.441 | 0.469 | 0.470 |
| | 336 | 0.464 | 0.439 | 0.451 | 0.443 | 0.484 | 0.458 | 0.487 | 0.458 | 0.497 | 0.445 | 0.546 | 0.496 | 0.570 | 0.546 | 0.493 | 0.487 | 0.565 | 0.515 | 0.491 | 0.469 | 0.489 | 0.467 | 0.530 | 0.499 |
| | 720 | 0.471 | 0.465 | 0.463 | 0.463 | 0.498 | 0.482 | 0.503 | 0.491 | 0.494 | 0.461 | 0.544 | 0.517 | 0.653 | 0.621 | 0.526 | 0.526 | 0.594 | 0.558 | 0.521 | 0.500 | 0.513 | 0.510 | 0.598 | 0.544 |
| | Avg | 0.434 | 0.427 | 0.426 | 0.431 | 0.447 | 0.440 | 0.454 | 0.447 | 0.455 | 0.429 | 0.507 | 0.472 | 0.529 | 0.522 | 0.475 | 0.481 | 0.541 | 0.507 | 0.458 | 0.450 | 0.461 | 0.458 | 0.498 | 0.484 |
| ETTh2 | 96 | 0.284 | 0.330 | 0.290 | 0.340 | 0.289 | 0.341 | 0.297 | 0.349 | 0.290 | 0.336 | 0.308 | 0.355 | 0.745 | 0.584 | 0.372 | 0.424 | 0.400 | 0.440 | 0.340 | 0.374 | 0.340 | 0.394 | 0.358 | 0.397 |
| | 192 | 0.357 | 0.376 | 0.379 | 0.396 | 0.372 | 0.392 | 0.380 | 0.400 | 0.372 | 0.385 | 0.393 | 0.405 | 0.877 | 0.656 | 0.492 | 0.492 | 0.528 | 0.509 | 0.402 | 0.414 | 0.482 | 0.479 | 0.429 | 0.439 |
| | 336 | 0.392 | 0.406 | 0.423 | 0.435 | 0.386 | 0.414 | 0.428 | 0.432 | 0.402 | 0.416 | 0.427 | 0.436 | 1.043 | 0.731 | 0.607 | 0.555 | 0.643 | 0.571 | 0.452 | 0.452 | 0.591 | 0.541 | 0.496 | 0.487 |
| | 720 | 0.391 | 0.418 | 0.473 | 0.465 | 0.412 | 0.434 | 0.427 | 0.445 | 0.430 | 0.444 | 0.436 | 0.450 | 1.104 | 0.763 | 0.824 | 0.655 | 0.874 | 0.679 | 0.462 | 0.468 | 0.839 | 0.661 | 0.463 | 0.474 |
| | Avg | 0.356 | 0.383 | 0.391 | 0.409 | 0.365 | 0.395 | 0.383 | 0.407 | 0.374 | 0.395 | 0.391 | 0.412 | 0.942 | 0.684 | 0.574 | 0.531 | 0.611 | 0.550 | 0.414 | 0.427 | 0.563 | 0.519 | 0.437 | 0.449 |
| ETTm1 | 96 | 0.313 | 0.342 | 0.329 | 0.366 | 0.320 | 0.357 | 0.334 | 0.368 | 0.318 | 0.349 | 0.352 | 0.374 | 0.404 | 0.426 | 0.365 | 0.387 | 0.364 | 0.387 | 0.338 | 0.375 | 0.346 | 0.374 | 0.379 | 0.419 |
| | 192 | 0.362 | 0.371 | 0.363 | 0.380 | 0.361 | 0.381 | 0.390 | 0.393 | 0.365 | 0.372 | 0.374 | 0.387 | 0.450 | 0.451 | 0.403 | 0.408 | 0.398 | 0.404 | 0.374 | 0.387 | 0.382 | 0.391 | 0.426 | 0.441 |
| | 336 | 0.385 | 0.389 | 0.390 | 0.404 | 0.390 | 0.404 | 0.426 | 0.420 | 0.401 | 0.397 | 0.421 | 0.414 | 0.532 | 0.515 | 0.436 | 0.431 | 0.428 | 0.425 | 0.410 | 0.411 | 0.415 | 0.415 | 0.445 | 0.459 |
| | 720 | 0.444 | 0.424 | 0.460 | 0.443 | 0.454 | 0.441 | 0.491 | 0.459 | 0.460 | 0.432 | 0.462 | 0.449 | 0.666 | 0.589 | 0.489 | 0.462 | 0.487 | 0.461 | 0.478 | 0.450 | 0.473 | 0.451 | 0.543 | 0.490 |
| | Avg | 0.376 | 0.382 | 0.386 | 0.398 | 0.381 | 0.396 | 0.410 | 0.410 | 0.382 | 0.386 | 0.402 | 0.406 | 0.513 | 0.495 | 0.423 | 0.422 | 0.419 | 0.419 | 0.400 | 0.406 | 0.404 | 0.408 | 0.448 | 0.452 |
| ETTm2 | 96 | 0.169 | 0.244 | 0.174 | 0.255 | 0.175 | 0.258 | 0.180 | 0.264 | 0.168 | 0.247 | 0.183 | 0.270 | 0.287 | 0.366 | 0.197 | 0.296 | 0.207 | 0.305 | 0.187 | 0.267 | 0.193 | 0.293 | 0.203 | 0.287 |
| | 192 | 0.233 | 0.288 | 0.239 | 0.299 | 0.237 | 0.299 | 0.250 | 0.309 | 0.234 | 0.291 | 0.255 | 0.314 | 0.414 | 0.492 | 0.284 | 0.361 | 0.290 | 0.364 | 0.249 | 0.309 | 0.284 | 0.361 | 0.269 | 0.328 |
| | 336 | 0.291 | 0.327 | 0.301 | 0.340 | 0.298 | 0.340 | 0.311 | 0.348 | 0.297 | 0.333 | 0.309 | 0.347 | 0.597 | 0.542 | 0.381 | 0.429 | 0.377 | 0.422 | 0.321 | 0.351 | 0.382 | 0.429 | 0.325 | 0.366 |
| | 720 | 0.387 | 0.387 | 0.395 | 0.396 | 0.391 | 0.396 | 0.412 | 0.407 | 0.386 | 0.385 | 0.412 | 0.404 | 1.730 | 1.042 | 0.549 | 0.522 | 0.558 | 0.524 | 0.408 | 0.403 | 0.558 | 0.525 | 0.421 | 0.415 |
| | Avg | 0.270 | 0.312 | 0.277 | 0.322 | 0.275 | 0.323 | 0.288 | 0.332 | 0.271 | 0.314 | 0.290 | 0.334 | 0.757 | 0.611 | 0.353 | 0.402 | 0.358 | 0.404 | 0.291 | 0.333 | 0.354 | 0.402 | 0.305 | 0.349 |

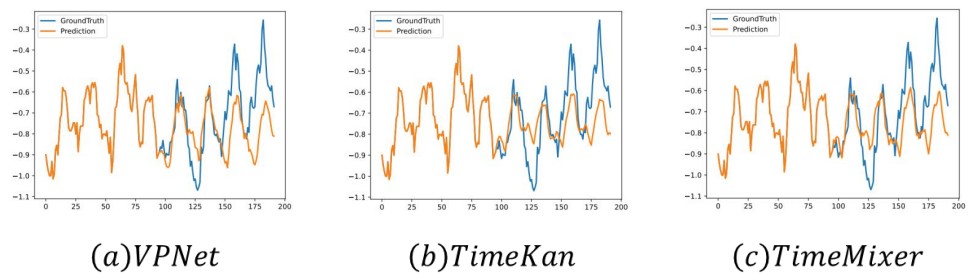

Figure 9: Showcases from ETTh1 by different models under the input-96-predict-96 settings.

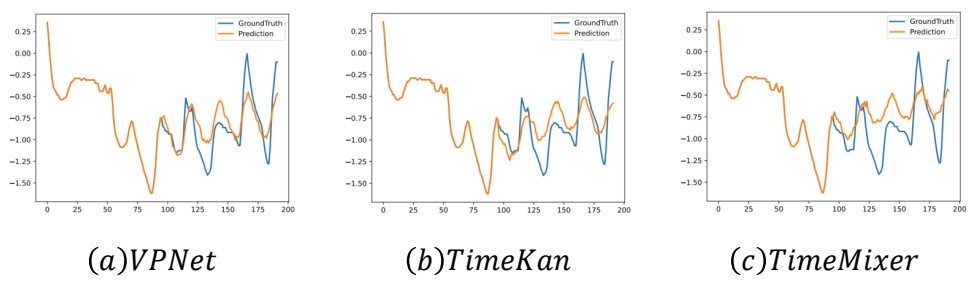

Figure 10: Showcases from ETTh2 by different models under the input-96-predict-96 settings.

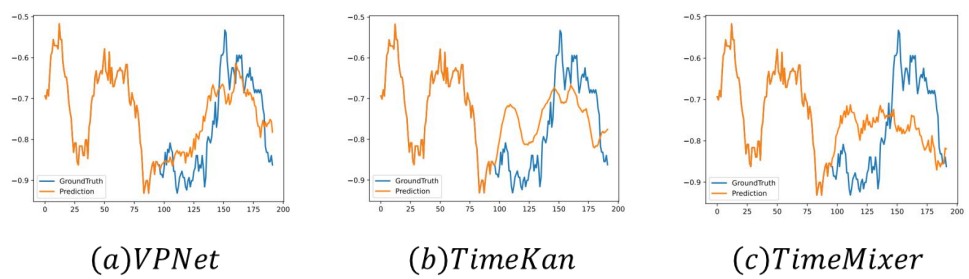

Figure 11: Showcases from ETTm1 by different models under the input-96-predict-96 settings.

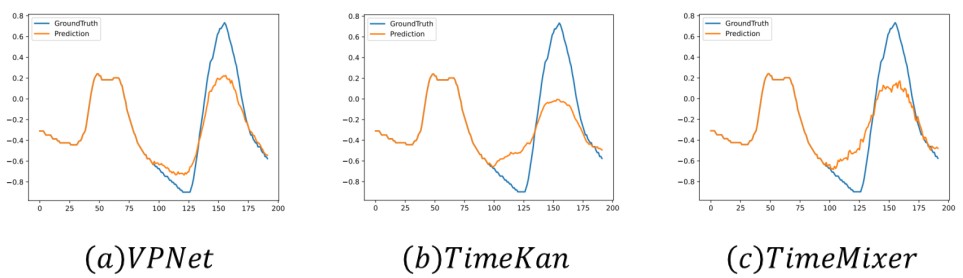

Figure 12: Showcases from ETTm2 by different models under the input-96-predict-96 settings.

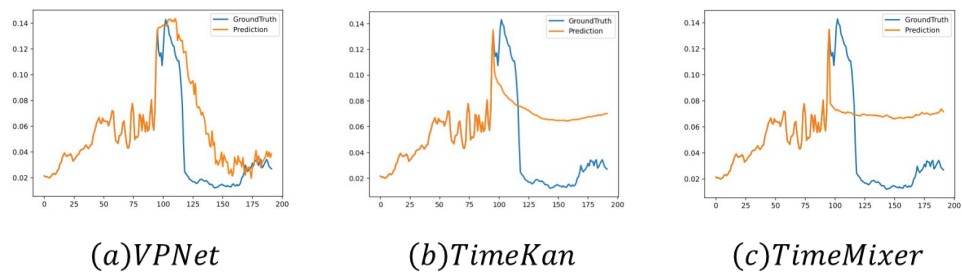

Figure 13: Showcases from Weather by different models under the input-96-predict-96 settings.

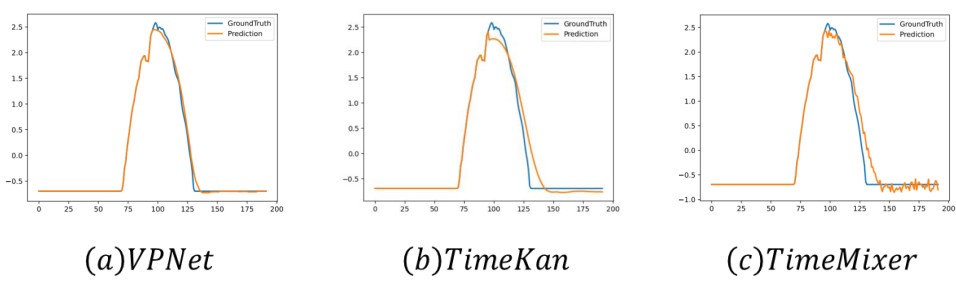

Figure 14: Showcases from Solar-Energy by different models under the input-96-predict-96 settings.

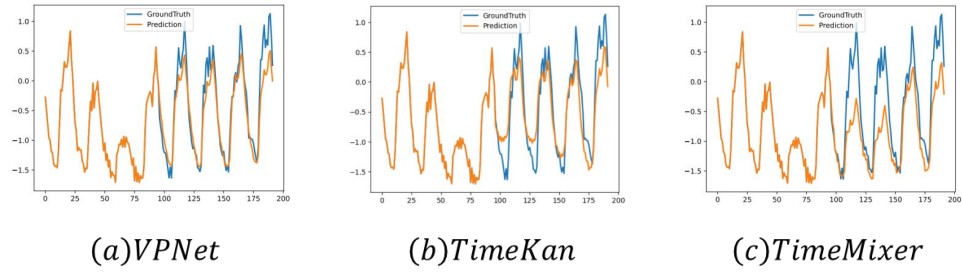

Figure 15: Showcases from Electricity by different models under the input-96-predict-96 settings.

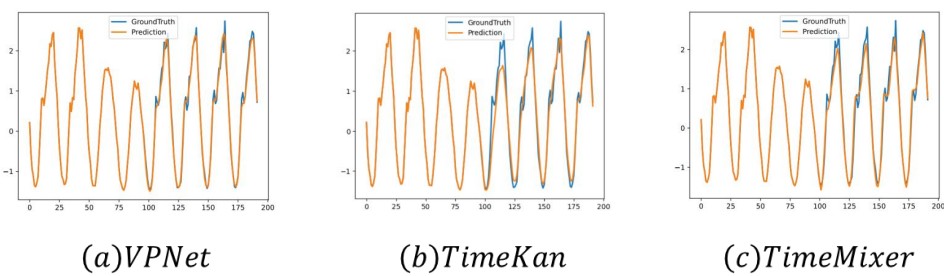

Figure 16: Showcases from Traffic by different models under the input-96-predict-96 settings.

