# OpenReview forum: "Are Global Dependencies Necessary? Scalable Time Series Forecasting via Local Cross-Variate Modeling"
_ICLR.cc/2026/Conference — ICLR 2026 Poster_

### Official Review · Reviewer_vhBd · 2025-10-30

**Soundness:** 2
**Presentation:** 2
**Contribution:** 3
**Rating:** 6
**Confidence:** 4

**Summary:**

This paper proposes VPNet, a multivariate time-series forecasting framework that models localized cross-variate interactions via depthwise 2D convolutions on a variate–patch representation. The authors argue that global dependency modeling is not necessary and introduce the Local Sufficiency Hypothesis as theoretical justification. Experiments on eight public datasets show improvements over recent baselines.

**Strengths:**

1. The paper provides a clear and well-motivated perspective questioning the necessity of global dependency modeling in multivariate forecasting. By framing the problem through the Local Sufficiency Hypothesis, the work contributes a conceptual shift that has relevance for both theory and practical model design.
2. The proposed Variate–Patch Field representation is intuitive yet effective, allowing variable-wise and temporal patterns to be captured jointly using simple depthwise 2D convolutions. This design reduces computational complexity from quadratic to linear with respect to the number of variables while preserving modeling capacity.
3. The empirical evaluation is thorough, covering diverse datasets with significantly different dimensionalities and dynamics. The model consistently outperforms competitive baselines, particularly in high-dimensional settings where efficiency matters most.
Empirical results are strong and consistent across multiple datasets.

**Weaknesses:**

1. Insufficient empirical support for the Local Sufficiency Hypothesis.
The central assumption that cross-variate dependencies are predominantly local is not thoroughly validated using real-world correlation or interaction patterns. The justification relies more on theoretical reasoning than on dataset-driven evidence, which weakens the persuasiveness of the hypothesis.
2. Lack of discussion on scenarios where locality may be insufficient.
While the model performs well on the evaluated datasets, it is unclear how it behaves in settings with strong long-range or sparse inter-variable dependencies. Without an analysis of potential failure cases or applicability boundaries, the generality of the proposed approach remains uncertain.
3. Limited interpretability analysis regarding learned dependency structure.
Given that the core contribution concerns how the model captures local cross-variate relationships, the paper would benefit from visualizations or interpretability studies (e.g., patch importance, receptive field inspection). These insights are necessary to substantiate the claims about the model’s learned dependency patterns.

**Questions:**

1. The Local Sufficiency Hypothesis is motivated theoretically under random or dense-dependency assumptions.  Could the authors provide empirical analyses of variable correlation / Granger causality / covariance decay across datasets to verify that real-world dependencies are indeed predominantly local?
2. Can the authors identify conditions under which VPNet fails or requires adaptation? A discussion on when local modeling is insufficient would help clarify the scope of applicability.
3. Since VPNet essentially enforces locality through convolutional receptive fields, how does it compare to local-window attention or strided/sparse attention variants on the same datasets?
4. The model claims to capture local cross-variate structure. Can the authors provide patch-level saliency / variable importance / receptive field visualization to show what dependencies the model actually learns?
5. The patch autoencoder is claimed to improve robustness to non-stationarity. Do the authors have shifted-train/test evaluations (e.g., seasonal shift, regime shift in traffic/energy data) to demonstrate this effect explicitly?

---

> ### Author Response · Authors · 2025-11-24
> **Response to Reviewer-vhBd (Part 1/2)**
>
> We sincerely thank the reviewer-vhBd for the rigorous assessment and valuable suggestions, especially the call to strengthen data-driven empirical validation beyond theoretical arguments and to further analyze the model's applicability boundaries and interpretability. To address these concerns, we have added the corresponding analyses and will provide detailed point-by-point responses based on these new results.
>
> > **W1 & Q1: The Local Sufficiency Hypothesis is motivated theoretically under random or dense-dependency assumptions. Could the authors provide empirical analyses of variate correlation / Granger causality / covariance decay across datasets to verify that real-world dependencies are indeed predominantly local?**
> >
>
> We sincerely appreciate the reviewer's insightful suggestion. To verify that real-world time-series datasets indeed exhibit predominantly dense dependencies, and to provide empirical support for the Local Sufficiency Hypothesis, we conducted a comprehensive statistical analysis across all benchmark datasets (see table below).
> Following the reviewer's guidance, we evaluated three key indicators of dependency density:
>
> 1. **Information redundancy (covariance decay).**
> Measured using PCA. We report the effective dimensionality reduction ratio (total variates / number of principal components explaining 95% variance) and the variance explained by the first principal component (PC1 Var). Faster decay indicates higher redundancy.
> 2. **Topological density (Granger causality).**
> After removing global-mean trends and differencing to avoid spurious regression, we compute the proportion of significant causal pairs (p<0.01, lag=3).
> 3. **Correlation strength.**
> Assessed using the distribution of pairwise Pearson correlations.
>
> **Conclusion:** Our results show that mainstream forecasting datasets predominantly fall into _dense-dependency scenarios_, likely because they are collected from tightly coupled physical or operational systems. For example, the Electricity dataset exhibits extremely high Granger causality density (89.19%) and strong redundancy (PC1 explains more than 50% of total variance). These empirical results confirm that cross-variate dependencies are widespread and redundant, meaning that local neighborhoods naturally contain sufficient information to recover global dynamics. This provides concrete evidence supporting the Local Sufficiency Hypothesis.
>
> **Table: Empirical Analysis of Multivariate Time-Series:** Mainstream Datasets Exhibit a Dense and Redundant Regime (Details in Appendix D)
>
> | **Dataset** | **Granger Density** | **PCA: PC1 Var** | **PCA: 95% Comps** | **PCA: Comp. Ratio** | **Corr: Avg \|r\|** | **Corr > 0.3** | **Corr > 0.5** | **Corr > 0.7** |
> | --- | --- | --- | --- | --- | --- | --- | --- | --- |
> | **Solar** | 51.87% | **91.77%** | **4** / 136 | **34.00x** | **0.9167** | **100.0%** | **100.0%** | **100.0%** |
> | **Traffic** | 69.45% | 57.67% | 202 / 862 | 4.27x | 0.5638 | 89.92% | 66.84% | 25.67% |
> | **Electricity** | **89.19%** | 54.68% | 86 / 321 | 3.73x | 0.4893 | 68.29% | 46.37% | 32.49% |
> | **Weather** | 64.52% | 42.44% | 9 / 21 | 2.33x | 0.2956 | 35.71% | 25.24% | 20.95% |
> | **ETTh1** | 64.29% | 34.39% | 5 / 7 | 1.40x | 0.2221 | 19.05% | 9.52% | 9.52% |
> | **ETTh2** | 52.38% | 43.10% | 5 / 7 | 1.40x | 0.3246 | 42.86% | 28.57% | 4.76% |
> | **ETTm1** | 47.62% | 34.57% | 5 / 7 | 1.40x | 0.2243 | 19.05% | 9.52% | 9.52% |
> | **ETTm2** | 30.95% | 43.08% | 5 / 7 | 1.40x | 0.3245 | 42.86% | 28.57% | 4.76% |
>
>
>
>
> > **W2 & Q2: Can the authors identify conditions under which VPNet fails or requires adaptation? A discussion on when local modeling is insufficient would help clarify the scope of applicability.**
> >
> We thank the reviewer for raising this important question. As stated in the paper, VPNet is best suited for **dense-dependency scenarios**, which cover the majority of widely used benchmarks (see Appendix D). In these settings, cross-variate dependencies are widespread and redundant, so a finite local neighborhood is very likely to contain informative neighbors, and local modeling is both effective and efficient.
>
> In contrast, local modeling can become insufficient in **sparse-dependency scenarios**. Our synthetic stress tests in Appendices E and F approximate such cases and indeed show that VPNet can degrade under random variate orderings. In such sparse-dependency scenarios, additional care is needed when applying VPNet, as its performance may degrade under random variate ordering. We provide two practical strategies to preserve its competitiveness in these cases:
>
> 1. Increasing the kernel size to expand the receptive field.
> 2. Applying correlation-based variate ordering to enrich local neighborhoods with informative signals.

---

> ### Author Response · Authors · 2025-11-24
> **Response to Reviewer-vhBd (Part 2/2)**
>
> > **Q3: Since VPNet essentially enforces locality through convolutional receptive fields, how does it compare to local-window attention or strided/sparse attention variants on the same datasets?**
> >
>
> We thank the reviewer for the suggestion to compare our convolution-based approach with local attention variants. To address this, we implemented two Transformer-based baselines: LANet (local-window attention) and SANet (sparse attention). We evaluated them on all eight benchmark datasets (see Table 1) as well as on the constructed Hetero-Mix dataset, which concatenates Traffic, Electricity, and Weather to contrast dense- versus sparse-dependency settings (see Table 2).
>
> Our results show that in dense-dependency scenarios with unknown variate ordering (e.g., Traffic and Electricity), VPNet consistently outperforms the attention-based variants. The strong inductive bias of TCN acts as an effective regularizer and provides more stable optimization than data-driven attention weights. Notably, in sparse-dependency settings, correlation-based ordering allows the local attention variant (LANet) to achieve performance comparable to VPNet. This suggests that attention mechanisms can reach a higher theoretical ceiling when a reliable structural ordering is available.
>
> **Table 1: Average Performance Comparison with Variants across 8 Datasets. (Details in Section 4.4)**
>
> | Model | Weather | Solar | Elec | Traffic | ETTh1 | ETTh2 | ETTm1 | ETTm2 |
> | :--- | :--- | :--- | :--- | :--- | :--- | :--- | :--- | :--- |
> | **VPNet** (TCN) | **0.238** | **0.204** | **0.162** | **0.421** | **0.434** | **0.356** | **0.376** | **0.270** |
> | **LANet** (Local) | 0.250 | 0.205 | 0.171 | 0.424 | 0.440 | 0.363 | 0.384 | 0.278 |
> | **SANet** (Sparse) | 0.252 | 0.205 | 0.173 | 0.425 | 0.440 | 0.361 | 0.384 | 0.277 |
>
>
> **Table 2: Variate-Ordering Experiments on Hetero-Mix (Appendix E).**
> Hetero-Mix simulates a sparse-dependency scenario that is uncommon in real forecasting tasks. It is created by concatenating the Traffic, Electricity, and Weather datasets along the channel dimension.
>
> | Ordering Setting | VPNet (Avg MSE) | LANet (Avg MSE) |
> | --- | --- | --- |
> | Hetero-Mix (Ordered) | **0.385** | **0.384** |
> | Hetero-Mix (Shuffled) | **0.388** | 0.392 |
>
>
> > **W3 & Q4: The model claims to capture local cross-variate structure. Can the authors provide patch-level saliency / variate importance / receptive field visualization to show what dependencies the model actually learns?**
> >
>
>
>
> We thank the reviewer for the thoughtful suggestion. To illustrate the dependencies learned by VPNet, we performed gradient-based saliency analysis (see **Figures 5–7 in Appendix G**). As shown in Figures 5–7, we visualized the receptive field when predicting the first future step of representative variates across different datasets:
>
> + **ETTh1 (variate 2):** The saliency map shows a strongly autoregressive pattern. Gradients concentrate almost entirely on the variate's own history, with very limited contribution from neighboring variates.
> + **Electricity (variate 200):** The model reveals meaningful cross-variate interactions. Notably, the prediction of variate 200 relies heavily on the history of variate 182, indicating that VPNet can detect specific inter-variate relationships within its receptive field.
> + **Traffic (variate 400):** The model activates a wide range of nearby variates when predicting the first step. This suggests a **collective neighborhood influence**, where VPNet aggregates the joint state of local variate clusters to infer future behavior.
>
>
>
> > **Q5: The patch autoencoder is claimed to improve robustness to non-stationarity. Do the authors have shifted-train/test evaluations (e.g., seasonal shift, regime shift in traffic/energy data) to demonstrate this effect explicitly?**
> >
> We would like to clarify that the patch autoencoder is adopted from existing state-of-the-art frameworks (AdaPatch [Liu et al., 2025]) as stated in our **PRELIMINARIES** section. It serves as the backbone component but is **not a novel contribution** of this paper. In this work, we did not perform additional shifted train/test evaluations specifically targeting non-stationarity beyond what was already reported in the AdaPatch framework. Since the patch autoencoder is not our main contribution, we chose to focus our experimental budget on validating VPNet’s locality assumptions.
>
> Reference:
> AdaPatch: Adaptive Patch-Level Modeling for Non-Stationary Time Series Forecasting.

---

> > ### Comment · Reviewer_vhBd · 2025-11-27
> >
> > The additional analyses clarify the dense-dependency conditions under which the proposed method is most effective; however, they do not fully establish the general validity of the Local Sufficiency Hypothesis. Static dependency measures alone are insufficient to guarantee predictive local sufficiency, and the model’s sensitivity to variate ordering remains a concern. The proposed strategies for sparse settings are reasonable but partially weaken the original locality-based motivation. Given these remaining limitations of the underlying assumptions, my overall score remains unchanged.

---

> ### Author Response · Authors · 2025-11-27
>
> Dear Reviewer-vhBd,
>
> We sincerely thank the reviewer for the constructive and insightful comments. **As we state in the introduction**, our work is driven by the fundamental question: “Is searching for global dependencies necessary for accurate forecasting in dense, high-dimensional systems?” We appreciate the reviewer’s acknowledgement of our Local Sufficiency Hypothesis under dense-dependency conditions, as well as the suggestion to consider broader settings. However, such broader scenarios are relatively **uncommon in current mainstream multivariate time-series benchmarks**, which predominantly arise from densely coupled systems. Motivated by the practical efficiency–accuracy trade-off in these regimes, we designed VPNet, which leverages a **higher-level variate–patch field** to achieve state-of-the-art performance while maintaining linear complexity with respect to the number of variates on high-dimensional datasets.
>
> We agree that introducing learnable or data-driven variate ordering is a promising direction to further extend our approach to a wider range of scenarios. We regard this as an exciting avenue for future work, but it is **beyond the scope of the present paper**.
>
> We again thank the reviewer for the valuable feedback and for the time and effort devoted to evaluating our manuscript.
>
> Best regards,
>
> The Authors

---

### Official Review · Reviewer_XMH1 · 2025-11-01

**Soundness:** 3
**Presentation:** 3
**Contribution:** 3
**Rating:** 6
**Confidence:** 3

**Summary:**

The paper challenges the common assumption that multivariate forecasting requires global cross-variate modeling. It proposes the Local Sufficiency Hypothesis—that, in dense dependency regimes, a small local neighborhood across variates is enough for accurate prediction. Building on this, the authors introduce VPNet, which (i) encodes sequences into patch embeddings with a lightweight overcomplete autoencoder, (ii) channelizes them into a 2D variate–patch field, and (iii) processes this field with stacked VarTCNBlocks, yielding linear complexity in the number of variates. Extensive experiments on eight benchmarks report state-of-the-art (SOTA) or competitive results with strong efficiency.

**Strengths:**

1. The Local Sufficiency Hypothesis is a useful lens to revisit the global-vs-local trade-off, with a simple probabilistic argument that motivates local kernels.
2. The variate–patch field + VarTCNBlocks pipeline is conceptually clean and easy to implement. The channelization step makes the locality idea actionable.
3. The proposed design keeps computation linear in the number of variates while still modeling cross-variate signals—addressing a pain point of global attention models.
4. Evaluation spans 8 common datasets (Weather, Traffic, Electricity, Solar-Energy, ETTh1/2, ETTm1/2) with standardized horizons and protocol; results are competitive or SOTA.

**Weaknesses:**

1. The theory assumes random variate permutations when quantifying the probability that a local window contains an informative neighbor. Real systems often have structured (and non-exchangeable) dependencies; the proof does not address such structure nor provide guarantees beyond dense regimes.
2. While the paper claims robustness to different orderings in one setting, the practical definition of “local neighbors” depends on variate ordering. More systematic analysis (beyond a small-kernel, shallow-stack case) is needed, including learned or data-driven orderings and lag-aware neighborhoods.
3. The strongest gains appear on high-dimensional datasets (Electricity/Traffic). On lower-dimensional ETT, improvements are smaller and sometimes only “competitive,” suggesting locality helps most in dense regimes; the paper could better quantify when global modeling is still useful.

**Questions:**

1. Can VPNet **learn** variate orderings or local neighborhoods (e.g., via learned permutations, dynamic kernels, or dilation) rather than relying on fixed layouts?
2. Do the authors have stress tests where inter-series dependencies are sparse/heterophilous or dominated by long-range cross-variate relations?
3. The paper gives a probabilistic guideline for kernel selection; can the authors show empirical calibration of that rule (Eq. 20) on real datasets?

---

> ### Author Response · Authors · 2025-11-24
> **Response to Reviewer-XMH1 (Part 1/3)**
>
> We sincerely appreciate the reviewer-XMH1's constructive comments, especially the concerns regarding variate ordering. To address this, we analyzed _mainstream time series datasets_ and found that most of them correspond to **dense-dependency scenarios (see supporting experimental results)**, as time series are typically collected from tightly coupled systems. We acknowledge that **sparse-dependency scenarios** do exist, though they are less common. Using synthetic datasets, we further evaluated VPNet under such settings and showed that **correlation-based ordering** can effectively improve model performance (please refer to Appendix E and F). While structure-aware orderings are theoretically optimal, discovering such orderings is a complex problem. In dense-dependency scenarios, the additional complexity is generally unnecessary and would compromise VPNet's efficiency.
>
> Below, we provide detailed point-by-point responses.
>
> > **W1: The theory assumes random variate permutations when quantifying the probability that a local window contains an informative neighbor. Real systems often have structured (and non-exchangeable) dependencies; the proof does not address such structure nor provide guarantees beyond dense regimes.**
> >
>
> We clarify that the "random permutation" assumption serves as a **theoretical lower bound** for our performance guarantee. Our theory aims to demonstrate that in dense-dependency scenarios, VPNet remains effective even when the variate ordering is unknown or completely shuffled. Once the probability of capturing informative neighbors is guaranteed under random permutations, any ordering that groups correlated variates will **strictly increase** this probability.
>
> To further address the reviewer's concern about sparse-dependency scenarios, we added an experiment using a synthetic sparse benchmark.
>
> **Table: Variate-Ordering Experiments on Hetero-Mix (Appendix E).**
> Hetero-Mix simulates a sparse-dependency scenario that is uncommon in real forecasting tasks. It is created by concatenating the Traffic, Electricity, and Weather datasets along the channel dimension. LANet is a VPNet variant that uses local-window attention. The results show that VPNet achieves clear improvements when the variate order preserves prior structure. VPNet also shows stronger robustness than the attention-based variant when the ordering is random, which further supports its stability.
>
> | Ordering Setting | VPNet (Avg MSE) | LANet (Avg MSE) |
> | --- | --- | --- |
> | Hetero-Mix (Ordered) | **0.385** | **0.384** |
> | Hetero-Mix (Shuffled) | **0.388** | 0.392 |
>
>
> These findings confirm that VPNet is stable in dense-dependency scenarios. They also show that correlation-based ordering can provide additional benefits when the dependency structure is sparse.
>
> ****
>
> > **Q1: Can VPNet learn variate orderings or local neighborhoods (e.g., via learned permutations, dynamic kernels, or dilation) rather than relying on fixed layouts?**
> >
>
> VPNet currently adopts **a fixed layout**, which is **a deliberate design choice made to prioritize computational efficiency**. Although incorporating learnable permutations or dynamic kernels is theoretically feasible, such mechanisms often introduce substantial computational overhead while providing diminishing returns, which runs counter to our design goals. We acknowledge that learnable variate ordering is an important direction in time-series forecasting, and it will be a key focus of our future work.
>
> ---
>
> **Supporting Experimental Results**
>
> **Table: Empirical Analysis of Multivariate Time-Series:** Mainstream Datasets Exhibit a Dense and Redundant Regime (Details in Appendix D)
>
> | **Dataset** | **Granger Density** | **PCA: PC1 Var** | **PCA: 95% Comps** | **PCA: Comp. Ratio** | **Corr: Avg \|r\|** | **Corr > 0.3** | **Corr > 0.5** | **Corr > 0.7** |
> | --- | --- | --- | --- | --- | --- | --- | --- | --- |
> | **Solar** | 51.87% | **91.77%** | **4** / 136 | **34.00x** | **0.9167** | **100.0%** | **100.0%** | **100.0%** |
> | **Traffic** | 69.45% | 57.67% | 202 / 862 | 4.27x | 0.5638 | 89.92% | 66.84% | 25.67% |
> | **Electricity** | **89.19%** | 54.68% | 86 / 321 | 3.73x | 0.4893 | 68.29% | 46.37% | 32.49% |
> | **Weather** | 64.52% | 42.44% | 9 / 21 | 2.33x | 0.2956 | 35.71% | 25.24% | 20.95% |
> | **ETTh1** | 64.29% | 34.39% | 5 / 7 | 1.40x | 0.2221 | 19.05% | 9.52% | 9.52% |
> | **ETTh2** | 52.38% | 43.10% | 5 / 7 | 1.40x | 0.3246 | 42.86% | 28.57% | 4.76% |
> | **ETTm1** | 47.62% | 34.57% | 5 / 7 | 1.40x | 0.2243 | 19.05% | 9.52% | 9.52% |
> | **ETTm2** | 30.95% | 43.08% | 5 / 7 | 1.40x | 0.3245 | 42.86% | 28.57% | 4.76% |

---

> ### Author Response · Authors · 2025-11-24
> **Response to Reviewer-XMH1 (Part 2/3)**
>
> > **W2: While the paper claims robustness to different orderings in one setting, the practical definition of “local neighbors" depends on variate ordering. More systematic analysis (beyond a small-kernel, shallow-stack case) is needed, including learned or data-driven orderings and lag-aware neighborhoods.**
> >
>
>  We sincerely appreciate the reviewer's suggestion regarding the systematic analysis and optimization of variate ordering.  We acknowledge that, under the current architecture, the practical definition of “neighbors" does depend on the variate order. While learnable or data-driven orderings and lag-aware neighborhood structures can indeed improve performance, they also introduce substantial computational cost and yield diminishing returns in dense-dependency scenarios.
>
> **1.** To address the request for analysis beyond the “small-kernel, shallow-stack" setting, we clarify that this configuration was intentionally chosen as a **strict lower-bound stress test** of the Local Sufficiency Hypothesis.
> Shallow models with limited receptive fields represent the most vulnerable configuration with respect to ordering changes, since they lack global routing capacity. Our experiments confirm that even under these challenging settings, the performance gap among random, natural, and correlation-based orders remains statistically negligible on standard benchmarks. This strongly supports VPNet's robustness in dense-dependency scenarios. We will include expanded analysis on this topic in the next revision.
>
> **2.** We agree that in sparse-dependency scenarios, **end-to-end learnable variate ordering** can be crucial for achieving higher performance. However, such mechanisms tend to be computationally expensive and are often “theoretically ideal but practically costly."
> Given this trade-off, we do not integrate dynamic ordering modules in the current version. Instead, we adopt correlation-based data-driven ordering as a practical and efficient proxy. Our experiments show that this strategy effectively restores performance in sparse settings (please refer to Appendix E and F).
> We also recognize the value of dynamic, learnable ordering and have identified it as a promising direction for future research.
>
>
>
> > **Q2: Do the authors have stress tests where inter-series dependencies are sparse/heterophilous or dominated by long-range cross-variate relations?**
> >
>
> We thank the reviewer for the valuable suggestion. We have added a new stress test (see Appendix F) by injecting Gaussian noise into the Electricity dataset, creating a sparse dependency, low SNR environment. As shown in Table below, standard VPNet performs slightly worse under random ordering in this setting. However, we observe that increasing the kernel size allows VPNet to recover performance comparable to global-dependency models such as ITransformer. In addition, applying correlation-based ordering also leads to substantial performance gains.
>
> **Table: Stress test on Electricity + 10× noise (details in Appendix F).**
> This setup simulates a sparse, low-SNR scenario, which is relatively rare in time-series forecasting, where local signals are severely weakened.
>
> | Model | Ordered (Kernel = 27) | Shuffled (Kernel = 27) |
> | --- | --- | --- |
> | VPNet (Avg MSE) | **0.166** | 0.178 |

---

> ### Author Response · Authors · 2025-11-24
> **Response to Reviewer-XMH1 (Part 3/3)**
>
> > **W3: The strongest gains appear on high-dimensional datasets (Electricity/Traffic). On lower-dimensional ETT, improvements are smaller and sometimes only “competitive," suggesting locality helps most in dense regimes; the paper could better quantify when global modeling is still useful.**
> >
>
> We sincerely appreciate the reviewer's insightful suggestion. The effectiveness of local modeling indeed varies with the characteristics of the data. As noted,  the performance gains of VPNet on low-dimensional datasets such as ETT are indeed more limited, primarily because these datasets are low-dimensional and exhibit relatively sparse dependencies. Based on our new empirical analyses (Appendix D) and stress tests (Appendices E and F), we provide the following guidelines on **when global modeling or local modeling is more advantageous**:
>
> **Dense-dependency scenarios:** Datasets such as Traffic and Electricity fall into this category. They exhibit high redundancy (Granger density > 89%, compression ratio > 3.7×) and pervasive cross-variate correlations. In such settings, **VPNet performs better and is significantly more efficient**. Its strong inductive bias allows it to capture dense local structures effectively while avoiding the overfitting risk and computational overhead often associated with global attention mechanisms on dense datasets. Therefore, **VPNet is the preferred choice in such dense-dependency scenarios.
>
> **Sparse-dependency scenarios:** In these relatively rare scenarios, we recommend selecting the modeling strategy based on the dimensionality of the variates. For low-dimensional datasets such as ETTh1, which exhibit relatively sparse cross-variate dependencies (compression ratio ≈ 1.4×, proportion of strong correlations < 10%), global modeling approaches (e.g., iTransformer) are theoretically more suitable. Nevertheless, because the number of variates is small, VPNet can approximate a global receptive field by increasing the kernel size and thus still achieve competitive performance, making VPNet a suitable choice in this scenarios as well. For high-dimensional scenarios with sparse dependencies, global modeling provides a simple and effective way to obtain strong performance; however, when efficiency is a primary concern, we recommend applying correlation-based variate ordering to construct an “artificially” locally dense structure, thereby enabling VPNet to reach comparable performance.
>
> We will incorporate a more detailed analysis of these findings in the next version of the paper.
>
>
>
> > **Q3: The paper gives a probabilistic guideline for kernel selection; can the authors show empirical calibration of that rule (Eq. 20) on real datasets?**
> >
>
> We are happy to provide a concise empirical calibration of Eq. 20 using the Electricity dataset as a case study.
>
> We adopt the theoretical guideline
> $$ k \ge \frac{C-1}{r} \ln \frac{1}{\delta} \approx \frac{1}{Density} \ln \frac{1}{\delta} $$
> and use a strict confidence level of 99% ($\delta = 0.01, \ln(1/\delta)\approx 4.6$).
>
> On the Electricity dataset, we evaluate two notions of “density":
>
> 1. **Causal density (Granger density = 89.19%)**
>  This yields a lower bound of ($k \approx 6$). This value is consistent with our observations in Section 4.3, where even such small kernels already achieve performance comparable to global-dependency models (e.g., iTransformer; see Table below).
> 1. **Strong-correlation density (correlation > 0.5 = 46.37%)**
> This yields ($k \approx 10$), which aligns with our standard choice ($k=7$) that balances computational efficiency and model capacity.
>
> **Practical takeaway.**
> Minimal kernels (around 3–6) satisfy the lower-bound guarantee, but moderately larger kernels improve the probability of capturing strong predictors. We therefore recommend selecting $k$ based on strong-correlation density at high confidence, then scaling slightly upward to maximize information capacity while maintaining efficiency.
>
> **Table: Performance vs. Kernel Size on Electricity. (Details in Section 4.3).** Comparison with Global Attention Baseline (ITransformer).
>
> | Model | Kernel Size (k) | MSE (Electricity) | Relative to ITransformer |
> | :--- | :--- | :--- | :--- |
> | **ITransformer** (Global) | All (Global) | 0.178 | Baseline |
> | VPNet (Channel-Indep) | 1 | 0.184 | -3.3% |
> | **VPNet (Small Kernel)** | **3** | **0.171** | **+4.1% (Outperforms)** |
> | VPNet (Standard) | 7 | 0.167 | +6.6% |
> | VPNet (Large Kernel) | 17 | 0.162 | +9.8% |
> | **VPNet (Optimal)** | **27** | **0.160** | **+11.2%** |

---

> ### Comment · Reviewer_XMH1 · 2025-11-27
>
> Thank you for the response and new experiments. I will maintain my positive score.

---

> > ### Author Response · Authors · 2025-11-28
> >
> > Dear Reviewer-XMH1,
> >
> > We sincerely appreciate your positive evaluation and the time you have taken to carefully review our work and our response. If you have any further questions or suggestions, we would be very happy to discuss them. Thank you again for your time and helpful feedback.
> >
> >
> > Best regards,
> >
> > The Authors

---

### Official Review · Reviewer_WGdL · 2025-11-01

**Soundness:** 2
**Presentation:** 3
**Contribution:** 2
**Rating:** 4
**Confidence:** 4

**Summary:**

This paper tackles multivariate forecasting by questioning the need for global cross-variate attention and proposes VPNet, which targets local cross-variate interactions for better scalability. The authors offer theoretical and empirical support that dense dependency systems can be modeled effectively with locality. They also propose a specialized VarTCNBlock—depthwise 2D convolutions to capture joint spatio-temporal (cross-variate + temporal) structure, which haslinear complexity in the number of variates. Experiments across multiple datasets claim state-of-the-art accuracy with notable efficiency gains.

**Strengths:**

1. The motivation to reduce the complexity of capturing cross-variate dependencies is sound. The authors investigate whether we really need to capture global dependencies considering all variates.
2. The authors tried to provide some theoretical analysis for this local sufficiency hypothesis, which has some merits.
3. The efficiency of the proposed method seems good compared with other baselines, even iTransformer (which only considers cross-variate dependencies without explicitly considering time dependencies)

**Weaknesses:**

1. I feel that the theoretical analysis of the proposed Local Sufficiency Hypothesis is not very convincing. My concern is mainly on the orders of variates. Please see the later question I raise on the "finite local neighborhood".
2. The experiments are not comprehensive enough in my opinion. There are several more comprehensive benchmarks proposed since 2025, e.g., fev-benchmark [1] and Gift-Eval [2]. I would suggest to have some results on those benchmarks and investigate how it compare with other baselines.

References:

[1] fev-bench: A Realistic Benchmark for Time Series Forecasting

[2] GIFT-Eval: A Benchmark For General Time Series Forecasting Model Evaluation

**Questions:**

1. The title in the manuscript is different from that of the submission page. I think it should be revised.
2. I wonder how we define the "appropriately chosen finite local neighborhood" in Local Sufficiency Hypothesis. The variates naturally do not have orders and are permutation-invariant. Although the ablation study provides some results on the effect of variate reordering. I feel that might not be valid on all datasets. Some orders are better than others on some datasets. However, I do not observe some patterns/insights on the results - e.g., which order we should use for which dataset.
3. I am curious how performance may change if we vary the input length. Because in my view, the variate locality may heavily depend on the input length. For example, if we really have a short input length, it may not be able to capture the dependencies.

---

> ### Author Response · Authors · 2025-11-24
> **Response to Reviewer-WGdL (Part 1/3)**
>
> We appreciate the reviewer-WGdL's insightful comments regarding the "appropriately chosen finite local neighborhood" and the orders of variates.  We would like to clarify that VPNet's robustness to variate ordering is established within the **dense-dependency scenario**, which represents the predominant setting in the current time series forecasting domain (see supporting experimental results). In fact, **existing mainstream benchmarks can be categorized as dense-dependency scenarios**. However, we acknowledge that in the corresponding **sparse-dependency scenario**, VPNet's performance remains sensitive to variate ordering (please refer to Appendix E and F).
>
> Below is our point-by-point response.
>
> > **W1: I feel that the theoretical analysis of the proposed Local Sufficiency Hypothesis is not very convincing. My concern is mainly on the orders of variates. Please see the later question I raise on the "finite local neighborhood".**
> >
>
> _Please refer to our detailed response in __**Question 2 (Q2)**__ below, where we clarify the definition of "appropriately chosen finite local neighborhood" and the impact of variate ordering._
>
> > **W2: Comprehensiveness of experiments and suggestions on new benchmarks (fev-benchmark [1] and Gift-Eval [2]).**
> >
>
> We appreciate the reviewer for suggesting these **two newly proposed benchmarks**. We have carefully examined both and conducted additional experiments on `GIFT-Eval`.
>
> 1. **Regarding **`fev-bench`** [1]:** We investigated this benchmark and found it is primarily _designed for pretrained time-series foundation models_. It lacks a standard train-test split protocol suitable for training deep learning models from scratch (end-to-end). Therefore, it is mainly used for zero-shot evaluation and is not directly applicable to our experimental setting.
> 2. **Regarding **`GIFT-Eval`** [2]:** We investigated this benchmark and found that GIFT-Eval provides interfaces for both time-series foundation models and classical forecasting models, but the latter must be adapted to support probabilistic prediction. Following the adaptation code of ITransformer, we modified VPNet accordingly, integrated it into GIFT-Eval, and evaluated it against ITransformer. As shown in the table below, VPNet outperforms ITransformer on **17 out of 31 datasets** (MAPE), demonstrating its competitiveness while **maintaining much higher efficiency**. (More experimental details are provided in Appendix H.)
>
> **Table: Comparison on GIFT-Eval Benchmark (Metric: MAPE)**
>
> | **Dataset Setting** | **VPNet (Ours)** | **ITransformer** | **Winner** |
> | --- | --- | --- | --- |
> | **bizitobs_application/10S/long** | **0.054** | 0.065 | VPNet |
> | **bizitobs_application/10S/medium** | **0.043** | 0.047 | VPNet |
> | **bizitobs_application/10S/short** | **0.038** | 0.041 | VPNet |
> | **bizitobs_l2c/5T/long** | **0.463** | 0.537 | VPNet |
> | **bizitobs_l2c/5T/medium** | **0.461** | 0.567 | VPNet |
> | **bizitobs_l2c/5T/short** | **0.148** | 0.174 | VPNet |
> | **bizitobs_l2c/H/long** | **0.664** | 0.670 | VPNet |
> | **bizitobs_l2c/H/short** | **0.643** | 0.707 | VPNet |
> | **bizitobs_service/10S/short** | **0.147** | 0.161 | VPNet |
> | **ett1/15T/medium** | **0.735** | 0.749 | VPNet |
> | **ett2/15T/long** | **0.150** | 0.156 | VPNet |
> | **ett2/15T/medium** | **0.144** | 0.152 | VPNet |
> | **ett2/15T/short** | **0.139** | 0.162 | VPNet |
> | **ett2/D/short** | **0.368** | 0.464 | VPNet |
> | **ett2/H/short** | **0.138** | 0.139 | VPNet |
> | **ett2/W/short** | **0.159** | 0.292 | VPNet |
> | **jena_weather/D/short** | **0.887** | 1.120 | VPNet |
> | _(Other datasets)_ | _..._ | _..._ | _ITransformer_ |
>
>
> Overall, we again thank the reviewer for the valuable suggestions and for broadening our evaluation scope. The discussion of the newly suggested benchmarks, along with the additional GIFT-Eval results, has been added to **Appendix H** of the revised manuscript.

---

> ### Author Response · Authors · 2025-11-24
> **Response to Reviewer-WGdL (Part 2/3)**
>
> >  **Q1: The title in the manuscript is different from that of the submission page.**
> >
>
> Thanks for pointing out this oversight. We have corrected the title in the revised manuscript to match the submission page.
>
>
>
> >  **Q2: I wonder how we define the "appropriately chosen finite local neighborhood" in Local Sufficiency Hypothesis. The variates naturally do not have orders and are permutation-invariant. Although the ablation study provides some results on the effect of variate reordering.**
> >
>
> We thank the reviewer for this insightful question. We would like to **clarify** the definition of the "**appropriately chosen finite local neighborhood**" and provide the **analysis regarding the effect of variate ordering**.
>
> **1. Definition of "Appropriately Chosen":**
>
> The "appropriately chosen finite local neighborhood" is formally defined as the subset of variates covered by the receptive field that satisfies the information connectivity lower bound.
>
> According to our theoretical analysis in **Corollary C.2 (Eq. 20)**, to ensure that the model captures global dependencies with probability $1-\delta$, the receptive field size $k$ must satisfy:
>
> $$k \ge \frac{C-1}{r} \ln \frac{1}{\delta}$$
>
> where $C$ is the total number of variates and $r$ relates to the density of the system. Therefore, an "appropriately chosen" neighborhood is one where $k$ meets this threshold. In practice, satisfying this lower bound ensures that sufficient information flows through the mixing layer to recover global dynamics, regardless of the specific permutation of inputs.
>
> **2. The effect of variate ordering:**
>
> The reviewer's concern about the impact of variate ordering is insightful. However, we would like to emphasize that a model's sensitivity to ordering largely depends on the dependency density of the dataset. In dense-dependency scenarios, VPNet is robust to variate ordering, whereas in sparse-dependency scenarios, which are relatively rare in time-series forecasting, VPNet can be more affected by the specific ordering of variates. We support this conclusion using the ordering experiments on *eight widely adopted benchmark datasets reported in the original paper*, as well as *two additional experiments* included below.
>
>
>
> **Table 1: Variate-Ordering Experiments on Hetero-Mix (Appendix E)**. Hetero-Mix models a sparse-dependency scenario (_rare in time-series forecasting_) created by concatenating the benchmark datasets Traffic, Electricity, and Weather along the channel dimension. LANet is a VPNet variant implemented with local-window attention. We observe that VPNet outperforms the random order baseline when the variate order preserves prior structure. Notably, under random ordering VPNet exhibits greater robustness than the attention-based variant, further confirming its stability.
>
> | **Ordering Setting** | **VPNet (Avg MSE)** | **LANet (Avg MSE)** |
> | --- | --- | --- |
> | **Hetero-Mix (Ordered)** | **0.385** | **0.384** |
> | **Hetero-Mix (Shuffled)** | **0.388** | 0.392 |
>
>
> **Table 2: Stress test on Electricity + 10× noise (details in Appendix F).**
> This setup simulates a sparse, low-SNR scenario, which is relatively rare in time-series forecasting, where local signals are severely weakened.
>
> | Model | Ordered (Kernel = 27) | Shuffled (Kernel = 27) |
> | --- | --- | --- |
> | VPNet (Avg MSE) | **0.166** | 0.178 |
>
>
>
> >  **Q3: How performance may change if we vary the input length.**
> >
>
>  We thank the reviewer for highlighting the role of input length. Input length indeed influences both temporal dependency modeling and the manifestation of variate locality.
>
> + **Short input lengths:** As the reviewer noted, very short inputs restrict long-range temporal modeling. However, our _Local Sufficiency Hypothesis_ concerns variate locality. Even with limited temporal context, strong cross-variate correlations often remain, and VPNet effectively captures these instantaneous or short-term interactions.
> + **Long input lengths:** With longer inputs, the model's hierarchical structure naturally expands its effective receptive field, enabling it to capture broader temporal dependencies.
>
>  As shown by the sensitivity analysis on input length in **Appendix J, Figure 8** of the original paper, VPNet maintains stable performance relative to baseline models across different input lengths.

---

> ### Author Response · Authors · 2025-11-24
> **Response to Reviewer-WGdL (Part 3/3)**
>
> **Table: MSE comparison of VPNet, TimeMixer, and iTransformer across different input lengths on the Elec. and Traff. datasets (prediction horizon $= 96$).**
>
> | Input Length L | Elec. – VPNet | Elec. – TimeMixer | Elec. – iTransformer | Traff. – VPNet | Traff. – TimeMixer | Traff. – iTransformer |
> |----------------|---------------|--------------------|-----------------------|-----------------|---------------------|------------------------|
> | 48             | 0.169         | 0.201              | 0.178                 | 0.512           | 0.624               | 0.496                  |
> | 96             | 0.135         | 0.153              | 0.148                 | 0.384           | 0.487               | 0.396                  |
> | 144            | 0.127         | 0.139              | 0.134                 | 0.355           | 0.410               | 0.368                  |
> | 192            | 0.127         | 0.135              | 0.135                 | 0.350           | 0.405               | 0.365                  |
> | 240            | 0.129         | 0.136              | 0.134                 | 0.348           | 0.401               | 0.366                  |
> | 336            | 0.127         | 0.133              | 0.130                 | 0.341           | 0.387               | 0.361                  |
>
> ---
>
> **Supporting Experimental Results**
>
> **Table: Empirical Analysis of Multivariate Time-Series:** Mainstream Datasets Exhibit a Dense and Redundant Regime (Details in Appendix D)
>
> | **Dataset** | **Granger Density** | **PCA: PC1 Var** | **PCA: 95% Comps** | **PCA: Comp. Ratio** | **Corr: Avg \|r\|** | **Corr > 0.3** | **Corr > 0.5** | **Corr > 0.7** |
> | --- | --- | --- | --- | --- | --- | --- | --- | --- |
> | **Solar** | 51.87% | **91.77%** | **4** / 136 | **34.00x** | **0.9167** | **100.0%** | **100.0%** | **100.0%** |
> | **Traffic** | 69.45% | 57.67% | 202 / 862 | 4.27x | 0.5638 | 89.92% | 66.84% | 25.67% |
> | **Electricity** | **89.19%** | 54.68% | 86 / 321 | 3.73x | 0.4893 | 68.29% | 46.37% | 32.49% |
> | **Weather** | 64.52% | 42.44% | 9 / 21 | 2.33x | 0.2956 | 35.71% | 25.24% | 20.95% |
> | **ETTh1** | 64.29% | 34.39% | 5 / 7 | 1.40x | 0.2221 | 19.05% | 9.52% | 9.52% |
> | **ETTh2** | 52.38% | 43.10% | 5 / 7 | 1.40x | 0.3246 | 42.86% | 28.57% | 4.76% |
> | **ETTm1** | 47.62% | 34.57% | 5 / 7 | 1.40x | 0.2243 | 19.05% | 9.52% | 9.52% |
> | **ETTm2** | 30.95% | 43.08% | 5 / 7 | 1.40x | 0.3245 | 42.86% | 28.57% | 4.76% |

---

### Author Response · Authors · 2025-12-01
**Response to the Area Chair and Reviewers**

Dear Area Chair and Reviewers,
We sincerely thank you for the time and effort devoted to evaluating this work.

In this response, we first briefly restate the core contributions of our paper:

- We propose the **Local Sufficiency Hypothesis** and provide both theoretical and empirical evidence that in dense multivariate systems, fully global dependency modeling is not strictly necessary.
- We introduce the **variate–patch field** and the **VarTCNBlock**, a new representation and architecture that instantiate this hypothesis and enable efficient cross-variate dependency modeling with linear complexity in the number of variates.
- We establish **state-of-the-art performance** on eight diverse forecasting benchmarks, showing that VPNet can simultaneously achieve strong accuracy and linear scalability, effectively resolving the classical accuracy–efficiency trade-off in high-dimensional dense regimes.

At the same time, we acknowledge that some reviewers raised concerns about the potential limitations of our hypothesis and model in sparse-dependency scenarios, particularly that the variate ordering might lead to degraded VPNet performance.

We appreciate the reviewers' recognition of the Local Sufficiency Hypothesis under dense-dependency conditions, as well as the suggestion to consider broader settings. However, such broader scenarios are **relatively uncommon in current mainstream multivariate time series benchmarks**, which predominantly arise from densely coupled systems.

For sparser-dependency settings, we extend our analysis in the revised appendix with synthetic datasets that mimic these scenarios and derive practical guidelines for using VPNet:

- In **low-dimensional sparse settings**, VPNet can effectively approximate global modeling by increasing the TCN kernel size, which expands its receptive field.
- In **high-dimensional sparse settings**, we show that a simple correlation-based variate ordering can construct an "artificially local" structure that significantly improves performance. In the worst case, when no meaningful ordering exists (i.e., dependencies are extremely weak), VPNet gracefully degenerates toward channel-independent behavior, avoiding catastrophic performance degradation.

We agree with the reviewers that learnable or fully data-driven variate ordering is a promising direction to further extend VPNet beyond its original dense-regime focus, enabling high performance and efficiency in high-dimensional, sparse-dependency multivariate time series settings. We view this as an exciting avenue for future work, but it is beyond the scope of the present paper.

We again thank the area chair and reviewers for their time and effort.

Best regards,

The Authors

---

### Meta-Review · Area_Chair_LQ3g · 2025-12-20

**Summary:**

The reviewers generally found the paper to be well motivated and practically relevant, with a clear focus on improving scalability for high-dimensional multivariate forecasting. The main concerns centered on the validity and scope of the Local Sufficiency Hypothesis, especially the dependence on variate ordering and the lack of guarantees beyond dense-dependency regimes. Reviewers also asked for broader evaluation on newer benchmarks and for clearer empirical evidence supporting the dense-dependency premise.

**Reviewer Concerns:**

Several concerns were meaningfully addressed in the rebuttal. The authors expanded empirical support for the dense-dependency assumption through dataset-level statistical analyses, and they introduced stress tests for sparse-dependency settings, along with practical mitigation strategies. They also extended evaluation to GIFT-Eval and clarified why fev-bench is not directly compatible with the training-from-scratch setting, which partially addresses the “missing newer benchmarks” concern. Comparisons to local/sparse attention variants, as well as saliency-style analyses, help substantiate that the model captures cross-variational structure.

Some concerns remain outstanding, particularly that static dependency measures do not fully prove predictive “local sufficiency,” and that ordering sensitivity in sparse or structured regimes is not eliminated.

**Reviewer Scores:**

Reviewer XMH1 explicitly stated they will maintain a positive score, so no change is expected. Reviewer vhBd also stated that their score remains unchanged, despite acknowledging the new analyses as helpful, so the rating is likely to stay the same. Reviewer WGdL initially leaned slightly negative, but the rebuttal addresses the title issue, adds GIFT-Eval results, and provides evidence of kernel and input-length sensitivity, which directly addresses their main requests. Given these additions, a modest score increase from WGdL is plausible, for example, moving from slightly below threshold to around the borderline acceptance range.

---

### Decision · Program_Chairs · 2026-01-26

Accept (Poster)